# Ground-based MFRSR UV-Vis spectral retrievals of Saharan dust absorption at Izaña Observatory

Hiren Jethva<sup>1,2</sup>, Nick Krotkov<sup>2</sup>, Omar Torres<sup>2</sup>, Jungbin Mok<sup>3</sup>, Gordon Labow<sup>3</sup>, Elena Lind<sup>2</sup>, Tom Eck<sup>4,2</sup>, Wei Gao<sup>5</sup>, George Janson<sup>5</sup>, Scott Simpson<sup>5</sup>, Darrin Sharp<sup>5</sup>, Kathy Lantz<sup>6</sup>, Charles Wilson<sup>6</sup>, Africa Barreto<sup>7,8</sup>, Rosa García<sup>9,7</sup>, Sergey Korkin<sup>4,2</sup>, David Flittner<sup>10</sup>

<sup>1</sup>Morgan State University, Baltimore, MD, United States

<sup>2</sup>NASA Goddard Space Flight Center, Greenbelt, MD, United States

10 <sup>3</sup>Science Systems and Applications, Inc., Lanham, MD, United States

<sup>4</sup>University of Maryland, Baltimore County, GESTAR-II, Baltimore, MD, United States

<sup>5</sup>USDA UV-B Monitoring and Research Program, Natural Resource Ecology Laboratory, Colorado State University, Fort Collins, Colorado, USA

<sup>6</sup>Global Monitoring Laboratory (GML), Radiation, Aerosol, and Cloud Division (GRAD), NOAA, Boulder,

15 Colorado, USA

<sup>7</sup>Izaña Atmospheric Research Center (IARC), State Meteorological Agency (AEMET), Santa Cruz de Tenerife, Spain

<sup>8</sup>Group of Atmospheric Optics, University of Valladolid, Valladolid, Spain

<sup>9</sup>TRAGSATEC, Madrid, Spain

20 <sup>10</sup>NASA Langley Atmospheric Research Center, VA, United States

Correspondence to: Hiren Jethva (hiren.t.jethva@nasa.gov)

Abstract. This paper presents a multi-instrument synergistic technique to retrieve atmospheric dust aerosol columnar 25 effective imaginary refractive index (k), single scattering albedo (SSA), and absorption aerosol optical depth (AAOD). The technique combines: (a) aerosol information derived from the narrow field-of-view measurements by filter sunmoon-sky radiometer within the Aerosol Robotic Network (AERONET): spectral aerosol optical depth (AOD) and inversion properties; (b) the total, direct, and diffuse sky irradiance measurements from UV- and Vis-Multifilter Rotating Shadowband Radiometer (MFRSR); (c) trace gas columns from satellite measurements (OMI and OMPS). 30 The approach is demonstrated on the data collected at the Izaña Atmospheric Observatory (IZO), located at an altitude of 2.4 km on Tenerife Island, a unique site for Saharan dust column optical properties retrievals due to very clean background condition for calibrating the instrument. This multi-instrument synergy enables consistent column absorption retrievals from ultraviolet (UV) to visible (VIS) wavelengths, while effectively accounting separately for aerosol and gaseous (Ozone-O<sub>3</sub>, and Nitrogen Dioxide-NO<sub>2</sub>) absorption. The MFRSR calibration procedure relies on 35 observations acquired on cleaner days (AOD








inversions, mostly within ±0.03 for AOD>0.2, and ±0.02 at higher AOD (>0.4). This close correspondence confirms the consistency between the two fundamentally distinct inversion techniques and enhances confidence in the concurrent MFRSR UV wavelength inversions. We present a multi-year (2019-2023) MFRSR aerosol absorption record revealing enhanced dust absorption at UV wavelengths with noticeable intraseasonal and interannual variabilities, which are indicative of a varying composition of minerals (iron oxides) in the dust. The spectral aerosol absorption effects reduce the amount of surface-reaching UV radiation and slow down tropospheric photochemistry, which can have implications for air quality, human health, and ecosystem dynamics. The ongoing AERONET and MFRSR measurements currently made at the Santa Cruz ground-level site on Tenerife Island will continue to produce a unique, long-term ground-based UV spectral Saharan dust absorption dataset, providing a valuable reference for evaluating space-based UV aerosol absorption retrievals from instruments such as DSCOVR-EPIC, S5P-TROPOMI, and the most recently launched PACE-OCI. In addition to deriving spectral absorption properties, the enhanced sensitivity of UV measurements to the dust spectral absorption, demonstrated with the MFRSR inversion in this work, can be exploited for inferring the mineralogical composition of the dust aerosols, which is critical to improving the dust representation in Earth System Models.

# 1 Introduction

Light-absorbing aerosols, by attenuating incoming solar radiation, play a critical role in governing the magnitudes of radiative balance and tropospheric photochemistry, and influence local to regional air quality, thereby human health. In addition, aerosol-cloud interactions, both through in-situ microphysical processes (indirect effect) and radiative coupling (direct effect), further amplify their potential to alter the regional and global climate. The Intergovernmental Panel on Climate Change (IPCC) Sixth Assessment Report (AR6) emphasizes that aerosol absorption, particularly by substances like black carbon and mineral dust, is a significant source of uncertainty in understanding and modeling the Earth's climate system (IPCC 2021, Forster et al., 2021). This uncertainty arises from pronounced variability in aerosol optical and microphysical properties, driven by diverse compositions, as well as a limited availability of high-quality observational datasets.

Since its inception in 1993 at the NASA Goddard Space Flight Center, the ground-based Aerosol Robotic Network (AERONET) has grown into a global network spanning multiple continents and hundreds of sites, encompassing ecosystems affected by biomass burning, desert dust, and urban-industrial activity (Holben et al., 1998). AERONET employs an automated, sun-sky photometer for making direct sun light measurements across a range of wavelengths from ultraviolet (UV-340 nm and 380 nm) to the shortwave near-infrared (SWIR-1640 nm).

The exact combination of optical filters changed somewhat over the years, with the most recent instrument model (CIMEL CE318-T) having center wavelengths at 340, 380, 440, 500, 675, 870, 940, 1020, 1640 nm. Direct sun measurements for AOD determination are taken at all these wavelengths with the reported AOD at 340, 380, 440, 500, 675, 870, 1020, 1640 nm and water vapor column at 940 nm. Scattered sky radiance measurements are done at 380, 440, 500, 675, 870, 1020, 1640 nm (starting 2016) and with additional measurements at 340 nm starting in July 2025 after instrument calibration using CE318-T model and new operational instrument firmware. Current V3 AERONET








processing inversion algorithm applies only four key wavelengths (440, 670, 860, 1020 nm) to retrieve aerosol optical and microphysical properties, including, spectral single-scattering albedo (SSA), aerosol particle size distribution (PSD), and aerosol phase function (Dubovik et al., 2002; Sinyuk et al., 2020). Although AERONET makes AOD measurements at 340 nm and 380 nm wavelengths, the sky measurements, essential for aerosol absorption inversion, were made only at 380 nm starting 2016. Due to calibration uncertainties, however, 380-nm sky observations have not been used in the AERONET inversion. This leaves the near-UV spectral region currently devoid of aerosol absorption properties in the V3 database. However, plans are underway to incorporate the past (380 nm) and new (340, 380 nm) UV sky measurements in the upcoming AERONET V4 processing. Extraterrestrial "direct sun" signal calibration of the AERONET reference instruments ("sun masters") is done at high altitude remote sites (Izaña and Mauna Loa Atmospheric Observatories) using the Langley extrapolation analysis. This results in a very high AOD accuracy of ~0.002 in the visible and near infrared to ~0.009 in the UV (Eck et al., 1999) for the sun masters.

Since the launch of NASA's Earth Observation System (EOS: <a href="https://science.nasa.gov/earth-science/the-earth-observer/">https://science.nasa.gov/earth-science/the-earth-observer/</a>) program in the 1990s, tremendous progress has been made in characterizing the global distribution of aerosol optical and microphysical properties from a suite of satellite sensors having unique capabilities. These include active and passive sensors, multiangle imagers, and UV to VIS to SWIR spectral coverage. Among them, the Ozone Monitoring Instrument (OMI) on board the Aura satellite stands out as the only sensor to provide a two-decade long aerosol absorption record in the near-UV spectral region (Torres et al., 2007; Torres et al., 2018). The two-channel near-UV algorithm applied to OMI, and later extended to Deep Space Climate Observatory Earth Polychromatic Camera (DSCOVR-EPIC) taking global "snapshot" measurements of the entire sun-lit Earth Disc from Sun-Earth Lagrange 1 orbit every 1-2 hours and Low Earth Orbit Sentinel Precursor (S5P-TROPOMI) sensors, takes advantage of the enhanced sensitivity of the top-of-atmosphere (TOA) observations at UV wavelengths to aerosol absorption and retrieves AOD and SSA simultaneously at 388 nm. The evaluation of the SSA retrievals from these spaceborne sensors relied on extrapolation from the near-UV 388 nm to 440 nm to facilitate a direct comparison against AERONET inversion (Jethva et al., 2014; Torres et al., 2020; Ahn et al., 2021). The current lack of AERONET ground-based inversions of aerosol absorption at UV wavelengths restricts a direct evaluation at the satellite retrieval wavelength.

The SKYNET network, a ground-based radiometric observation system with sites across Asia and Europe, conducts automated spectral measurements of direct solar flux and sky radiances. These observations are used in inversion algorithms to derive spectral AOD and SSA across the near-UV to NIR spectral range (Nakajima et al., 1996; Hashimoto et al., 2012). The availability of the SKYNET SSA inversion dataset in the UV spectrum enabled, for the first time, a direct evaluation of OMI-derived SSA at UV wavelengths, thus eliminating the need for spectral extrapolation to visible wavelengths as required in AERONET-based evaluations (Jethva et al., 2019). However, a key limitation of SKYNET's inversion approach is its assumption of a fixed, spectrally neutral surface albedo (set at 0.1) across UV to visible wavelengths (Campanelli et al., 2015), which can lead to underestimation of SSA, particularly under low aerosol loading conditions (Mok et al., 2018). While SKYNET data proved valuable for







assessing OMI aerosol absorption products, the study was constrained to a limited number of sites, primarily in Asia and Europe.

The UV-Vis Multifilter Rotating Shadowband Radiometer (UV-VIS MFRSR) (Harrison et al., 1994) is a unique addition to the existing ground network for aerosol monitoring. The modified version of the MFRSR instrument makes 1-minute measurements of total hemispheric and diffuse hemispheric spectral irradiance measurements at multiple UV wavelengths (311, 325, 332, 340, 380 nm) and a single visible (440 nm) wavelength (Krotkov et al., 2005a; Mok et al., 2016, 2018). The SSA inversion algorithm developed and demonstrated by Krotkov et al. (2005b) uses these measurements, in synergy with the collocated direct AOD and inversion dataset from AERONET, to fit the observed diffuse-to-direct ratio measurements to the on-the-fly radiative transfer (RT) simulations to derive the imaginary part of the refractive index independently at these wavelengths. A brief description of past MFRSR field operations is provided in the following section.

In July 2019, the modified MFRSR (head # 582) was deployed at the Izaña Atmospheric Observatory (IZO) in Tenerife Island located at about 350 km northwest from the western coast of the Saharan Desert (see Figure 1). The instrument was in near-continuous operation until October 2023 and collected valuable spectral sky irradiance measurements at UV (311 nm through 380 nm) and visible (440 nm) wavelengths. In this work, we present a multi-year inversion record of UV-VIS spectral aerosol absorption retrieved from the MFRSR direct and diffuse irradiance measurements acquired at IZO. Section 2 describes the MFRSR instrument characteristics and calibration procedure. Section 3 presents the spectral absorption retrieval algorithm. Section 4 discusses the results, including the SSA retrievals at 440 nm and their comparison against those of AERONET, a multi-year inversion record of UV to VIS SSAs, and spectral characteristics of dust aerosol absorption. Section 5 summarizes the work and its future implications in the evaluation of satellite aerosol absorption retrievals.

### 2 MFRSR Instrument and Calibration Procedure

# 2.1 A BRIEF HISTORY OF THE MFRSR INSTRUMENT AND PAST OPERATIONS

The two UV-VIS MFRSR sensors were deployed during two separate field campaigns. First, during the September—October 2007 biomass-burning season in Santa Cruz, Bolivia, UV-MFRSR measurements revealed that brown carbon (BrC) exhibits enhanced UV absorption, which reduces available surface UV-B radiation (290-320 nm) and slows the rate of tropospheric photochemical reactions (Mok et al., 2016). This resulted in simulations of decreased ozone production (up to 18%) and reduced radical concentrations (OH, HO<sub>2</sub>, RO<sub>2</sub>), thereby mitigating some adverse effects of biomass-burning emissions. Second, the UV- and VIS-MFRSR were operated, alongside SKYNET and AERONET sensors, from April to August 2016 in Seoul, South Korea, to study urban pollution aerosols, including transported Asian dust (Mok et al., 2018). A combination of AERONET, MFRSR, and Pandora (AMP) retrievals provided aerosol absorption data across UV-B (305 nm) to NIR (870 nm) wavelengths, isolating aerosol and gaseous absorption (e.g., NO<sub>2</sub>, O<sub>3</sub>). These findings underscore the significance of multi-instrument analysis for measuring spectral aerosol absorption properties in the UV and VIS wavelengths for climate, photochemistry, and health studies.







The UV-VIS MFRSR instrument (head # 582) was installed at the Izaña Observatory, located at 2.4 km altitude above sea-level on Tenerife Island in summer 2019. The instrument was operated near-continuously until the end of 2023 and made total and diffuse irradiance measurements at discrete UV (305, 311, 325, 332, 340, and 380 nm) and one visible (440 nm) wavelength bands. Prior to the deployment at the Izaña Observatory, the MFRSR instrument was calibrated at the NOAA Central UV Calibration Facility (CUCF) to accurately characterize its radiometric, spectral and angular responses. Figure 2 shows the spectral response function, central wavelength (CW), wavelength range, and full-width-half-maximum (FWHM) of five bands of MFRSR head # 582 used in the present work. Table 1 lists the MFRSR and AERONET datasets employed in the present synergistic aerosol absorption algorithm.

#### 2.2 MFRSR ON-SITE CALIBRATION PROCEDURE

In this section, we describe a new, on-site MFRSR calibration procedure involving three steps that requires AERONET-measured spectral AODs, along with Rayleigh and trace gas absorption optical depths, to derive the top-of-atmosphere (TOA) calibration constant V<sub>0</sub> [milli-volts] that would have been measured by the MFRSR at the TOA. The improved calibration procedure at the IZO site has an additional step, compared to the previously adopted procedure for fine-mode particles. This second step corrects the direct and diffuse irradiances for the sun forward scattering (aureole) effect, which is necessary for large, coarse-mode particles (dust) and larger AOD.

# 2.2.1 Step 1: Calculations of Calibration Constant V<sub>0</sub> on Cleaner Days

The calibration procedure starts with identifying the mostly aerosol-free, cleaner days with an AERONET AOD (440 nm) < 0.1. The MFRSR instrument measures raw total (full hemispheric view) and diffuse (sun shadowed) voltages [milli-volts] near-simultaneously for seven narrow filter (solid state detector) channels every minute. There are 2 additional voltage measurements with the shadow band blocking the sky radiance in aureole directions on each side of the sun. These measurements are used to correct for the forward scattered solar light around the sun (aureole) (Harrison et al., 1994). The complete cycle takes ~10 sec and is repeated every 1 minute throughout the day. For onsite MFRSR calibration, we use spectrally and time interpolated AERONET level 1.5 extinction aerosol optical depths,  $\tau_{AER}(\lambda_{rad})$ , where  $\lambda_{rad}$  is the radiatively equivalent wavelength for each channel [Krotkov et al., 2005a]. The surface pressure adjusted scattering molecular optical depth (i.e., Rayleigh OD,  $\tau_{Ray}(\lambda_{rad})P$ ) and the absorption optical depths of trace gases (O<sub>3</sub> and NO<sub>2</sub>) are also computed using satellite or co-located ground-based (e.g., PANDORA) measurements. The total atmospheric column extinction optical depth is then multiplied with the direct-sun air-mass factor, sec(SZA), where SZA is solar zenith angle and combined with the cosine-corrected direct-normal voltage measurement,  $V_{dirn}^{Meas}/f_R$ , to calculate the calibration constant ( $V_0^{Clean}$ ) for each channel and measurement cycle throughout the day, as formulated in Equation (1):

$$\ln(V_0^{Clean}) = \ln(V_{dirn}^{Meas}/f_R) + \sec(SZA) \times \left[ \tau_{AER}(\lambda_{rad}) + \tau_{Rav}(\lambda_{rad})P + \tau_{aas}(\lambda_{rad}) \right]$$
(1)

 $\tau_{AER}$  is interpolated with a 2<sup>nd</sup> order polynomial from 4 AERONET direct sun level 1.5 AOD (340 nm, 380 nm, 440 nm, 500 nm) in log-log space to  $\lambda_{rad}$  (Eck et al. 1999);  $\tau_{Ray}$  is the Rayleigh optical depth adjusted to the atmospheric pressure,  $f_R$  is the laboratory-measured and cosine-normalized angular response (*i.e.*, see fig. 4 in Krotkov et al, 2005a), and  $\tau_{gas}$  is the combined optical depth of the trace gases of Ozone (O<sub>3</sub>) and NO<sub>2</sub> obtained from Aura/OMI




sensor for almost entire record, except for the last three months (July through September 2023) during which the total O<sub>3</sub> column densities retrieved from SNPP-OMPS were used. Additionally, the O<sub>3</sub> and NO<sub>2</sub> amounts measured from the collocated ground-based Pandora instrument were used, whenever available, during the same period. Note that all radiative transfer calculations accounting for the Rayleigh, gaseous, and aerosol atmospheres were carried out at the radiatively equivalent wavelength calculated for each measurement and spectral channel, as shown in Krotkov et al. (2005a). Figure 3 shows examples illustrating the Step 1 of the calibration procedure for clean days on August 20 and 31, 2020, when AERONET-measured AOD at 440 nm (AOD440) were less than 0.03. The true-color RGB image captured by the MODIS sensor onboard the Aqua satellite visually confirms no dust transport from the Saharan desert on these two dates. The natural logarithm of  $V_0$ ,  $ln(V_0)$ , calculated from equation (1) for each 1-minute measurement is shown on the left-hand y-axis as red-green-blue-black dots. The different colors show the effect of the iterative removal of outlier measurements with  $V_0$  values outside of  $3\sigma$  standard deviation. The final selections (182 and 184 retained measurements out of original 209 and 211) are shown as black dots, while the black horizontal lines show the daily average  $< ln(V_0) >$  values. The red lines show independent Langley intercept calibrations applied to all filtered MFRSR morning measurements between 8 and 12 UTC (black dots). There is excellent agreement with the AERONET-based calibration on August 31 when the AOD remained approximately constant. On August 20, the Langley intercept is 1% lower than the AERONET-based  $\langle ln(V_0) \rangle$  because the AOD systematically increased during the morning Langley calibration period. We do not use Langley calibrations in this work.

Equation (1) was applied to all 1-minute MFRSR morning measurement cycles between 8 and 12 UTC resulting in (after removing outliers using an iterative procedure that discards data points outside  $3\sigma$  standard deviation) a daily average  $< \ln(V_0^{Clean}) >$  value of 8.111 and 8.116 on the respective dates. An increase in  $V_0$  by 0.5% from August 20 to 31 is real and attributed to the reduction in the Sun-Earth distance between the two dates. The Step 1 procedure described above was applied to each MFRSR spectral channel for a total of the 223 cleanest days with an AOD<sub>440</sub><0.10 identified during the AERONET-MFRSR combined operation period 2019-2023.

#### 2.2.2 Step 2: Direct Voltage Correction on Dusty Days

Equation (1) can be inverted to calculate MFRSR aerosol optical depth: τ<sub>MFRSR</sub>(λ<sub>rad</sub>) assuming known daily average calibration constant (i.e., < ln(V<sub>0</sub><sup>Clean</sup>) >), which can be estimated either from Langley intercept (Harrison and Michalsky, 1994) or using our Step 1 calibration procedure. For days with low AOD or fine-mode dominated aerosols (e.g., smoke, Mok et al., 2016), the MFRSR-derived and AERONET measured AOD typically agree within estimated uncertainties, e.g., |τ<sub>AER</sub> - τ<sub>MFRSR</sub>| < 0.01 for AOD < 0.5 (Krotkov et al., 2005a). However, for dusty days dominated by coarse aerosols the V<sub>dirn</sub><sup>Meas</sup> becomes overestimated due to under corrected aureole effect. On the other hand, AERONET direct sun AOD measurements τ<sub>AER</sub>(λ<sub>rad</sub>) are much less affected by the dust aureole effect even for coarse dust particles due to CIMEL small field-of-view (~1.2°). Therefore, we calculate corrected direct-normal voltages, V<sub>dirn</sub><sup>Corr</sup> on dusty days with AERONET AOD<sub>440</sub> > 0.2 using equation (2) and the closest-in-time cleaner day calibration constant < V<sub>0</sub><sup>Clean</sup> >:

$$\ln(V_{dirn}^{Corr}) = \ln(\langle V_0^{Clean} \rangle) - \sec(SZA) \times \left[ \tau_{AER}(\lambda_{rad}) + \tau_{Rav}(\lambda_{rad}) P + \tau_{gas}(\lambda_{rad}) \right]$$
 (2)





A search procedure was used to find the closest cleaner day with AERONET AOD<sub>440</sub> < 0.10 by moving the time window by 1 day at a time in backwards and forwards steps. The procedure continued until a closest cleaner day is identified in either a forward or backward time step.

# 2.2.3 Step 3: Diffuse Voltage Correction on Dusty Days

In the final calibration step, the corrected diffuse voltage,  $V_{dif}^{Corr}$ , is calculated by subtracting the corrected direct horizontal voltage,  $V_{dirn}^{Corr}$ , from the cosine-corrected measured total voltage,  $V_{dirn}^{Meas}$ , as given in Equation 3:

$$V_{dif}^{Corr} = \frac{V_{Total}^{Meas}}{fT} - \cos(SZA)V_{dirn}^{Corr}$$
 (3)

Where  $f_T$  is the effective cosine correction factor applied to the total (direct plus diffuse) raw irradiance measurement (Krotkov et al., 2005a). Figure 4 illustrates the Step 2 and Step 3 of the calibration procedure for August 26, 2020. The three-step calibration procedure described above yields the corrected diffuse voltage,  $V_{dif}^{Corr}$  for dusty day observations. Note that while the partition between direct and diffuse irradiance is adjusted according to Eq. (2) and Eq. (3), the total voltage  $V_{Total}^{Meas}$  measured by MFRSR does not change in the calibration process. The dimensionless ratio of the corrected diffuse to corrected direct-normal irradiance is then used in the retrieval algorithm to retrieve the imaginary part of the complex refractive index for each 1-minute measurement on dusty days with AOD<sub>440</sub> > 0.2.

# 3 AEROSOL ABSORPTION INVERSION ALGORITHM

The MFRSR inversion procedure involves several steps, beginning with creating a matchup file of the temporally collocated AERONET direct and inversion measurements, simulating the phase matrix of spheroidal dust aerosols, performing RT simulations of downward surface-reaching direct and diffuse irradiance, and finally applying a direct fit between the measurements and simulations.

#### 3.1 DESCRIPTION OF THE RT MODEL

We employ a Gauss–Seidel radiative transfer model, formally referred to as the Arizona RT model, for simulating the at-surface spectral direct and diffuse irradiances. The 1-dimensional RT code accounts for gaseous absorption as well as molecular and aerosol multiple scattering (Herman and Browning, 1965). In this plane-parallel atmosphere, sphericity is included on the incoming solar beam, also known as the pseudo-spherical approximation (Caudill et al. 1997). The model creates a total of 100 atmospheric layers with associated molecular, aerosols, and input trace gases profiles. Absorption cross-sections of ozone and nitrogen dioxide are averaged for each channel using the appropriate bandwidths to derive effective values used in the RT model. The reflecting surface is modeled as Lambertian with albedo obtained from AERONET at 440 nm, which also is used at all MFRSR UV channels. The aerosol absorption inversion procedure begins with the availability of the AERONET Level 1.5 direct measurements of spectral AOD and inversion parameters of PSD and the real part of the refractive index at the Izaña site. Each instance of an MFRSR measurement was temporally collocated with the AERONET direct measurements within ±15 minutes and available








closest-in-time inversion products on a daily basis. Note that, no averaging was performed on the MFRSR measurements. Instead, all MFRSR measurements were assigned to the corresponding collocated AERONET parameters. For each instance of AERONET-MFRSR collocated points, the spectral AODs in the range 340-500 nm are used to derive a quadratic relationship between AOD and wavelengths in log-log space, which then was used to estimate the AOD at the MFRSR radiatively equivalent channel wavelengths.

# 3.2 SPHEROIDAL SHAPE TREATMENT OF DUST AEROSOLS

The Izaña Observatory in the Tenerife Island is located in the northwest pathways of long-range dust transport from the Sahara Desert. During such transport events, the mineral dust from the Sahara constitutes the major component of aerosols over the region. Sea salt and locally emitted aerosols might also contribute to the total loading although their contribution is minimal as evident from the very low AOD<sub>440</sub> (








Torres et al. (2018) simulated the phase matrix elements for the OMAERUV dust aerosol models. The aspect ratio distribution assumed in the OMAERUV calculations were taken from Dubovik et al. (2006), which is shown as the black curve in Figure 5.

Torres et al. (2018) demonstrated that using a near-UV aerosol lookup table (at 354 nm and 388 nm), based on phase matrix elements derived from spheroidal aspect ratios proposed by Dubovik et al. (2006), significantly reduced scan bias in AOD retrievals. This approach improved consistency between retrievals on the west side (characterized by lower backscattering or forward-scattering angles) and the east side (backscattering angles) of the OMI scan over the Saharan Desert. The improvements in the OMI AOD retrievals reflect the effectiveness of the assumed spheroidal aspect ratio distribution over the dust source region. A similar analysis for the Saharan dust transport region over the Atlantic Ocean after Torres et al. (2018), however, still showed discrepancies in the retrieved AOD between the two sides of the OMI scan. To address the issue further, four distinct sets of aspect ratio distributions were created empirically by gradually increasing the weighting factors for spherical particles (axis ratio=1.0). The aerosol look-up tables created using the four corresponding sets of phase matrix elements were used in the OMI aerosol algorithm to investigate the retrieval results. It was found that a specific aspect ratio distribution, shown as the red curve in Figure 5, produced near-consistent dust AOD retrievals on both sides of the OMI scan over the ocean. Unlike the aspect ratio distribution suggested by Dubovik et al. (2006), with a complete absence of spherical particles and bin values around it, the empirically derived axis ratio exhibits a continuous distribution, representing a mixture of spherical and spheroidal particles with higher weights given to the spherical particles, i.e., aspect ratio=1.0 and bin values around it.

The aspect ratio distribution of the Saharan mineral dust inferred from electronic microscope and in situ measurements as reported in the published literature shows a great deal of variability. Ried et al. (2003) sampled the African dust during the Puerto Rico Dust Experiment and found dust to have a significantly higher aspect ratio near 1.9 with standard deviation of 0.9 after long-range transport. Huang et al. (2020) compiled the aspect ratio of the mineral dust aerosols representative of the Saharan and Asian deserts published in the literature, and showed a trend of increasing aspect ratios for northern African dust after transatlantic advection. Specifically, the median aspect ratio increased from  $1.60 \pm 0.07$  in the dust source regions of North Africa to  $1.66 \pm 0.03$  for short-range transported dust and  $1.90 \pm 0.04$  for long-range transported dust. This increase in particle asphericity or aspect ratio during transport is attributed to the preferential settling of spherical particles, which have a greater terminal fall speed than aspherical particles of the same volume. Overall, the aspect ratios reported for North African dust were in the range 1.4-2.0.

More recently, Panta et al. (2023) reports high asphericity of dust particles collected during the FRontiers in dust minerAloGical coMposition and its Effects upoN climaTe (FRAGMENT) field campaign in the Moroccan Sahara in September 2019, with mean and median aspect ratios of ~1.55±0.38 and 1.46, noting that it does not vary strongly with particle diameter. Aryasree et al. (2024) found that the aspect ratio of the Saharan air layer measured at the Capo Verde and Caribbean sites did not change significantly and remained in the range 1.5-2.0.

Accurately representing the true aspect ratio distribution of dust airmass observed by the MFRSR during absorption inversion at the study site remains a challenging problem. As a first approximation, we adopted an empirically derived model previously used in satellite-based AOD retrievals from satellite UV instruments. It is important to note that








improved aerosol retrievals from satellite observations do not necessarily confirm the presence of a mixture of spherical and spheroidal particles. Rather, this improvement suggests that such a mixture best reproduces the observed solar backscattered UV (BUV) radiances, even though real dust particles are typically angular and irregular, not smooth and spheroidal as represented in the model. Driven by the improved OMI AOD retrievals over land and ocean using two distinct models of spheroidal aspect ratio distribution, the dust retrievals in the near-UV aerosol algorithm applied to OMI (Torres et al., 2018), S5p-TROPOMI (Torres et al., 2020), and DSCOVR-EPIC sensors (Ahn et al., 2021) was also updated.

Both AERONET and satellite retrieval algorithms assume spheroidal particle shapes primarily because: (1) computationally efficient scattering algorithms for irregularly shaped particles are not readily available, and (2) the spheroidal approximation captures some of the key scattering phase matrix angular features of randomly oriented desert dust particles reasonably well. More importantly, in the context of this study, incorporating these empirically derived aspect ratios into the aerosol LUT significantly improved the agreement between the retrieved SSA at 440 nm from the MFRSR and corresponding AERONET inversions. The results of this comparison are discussed in Section 4.1 and presented in Figure 9.

During the summer months, the Izaña observatory on Tenerife Island observes transported dust that originated in the Saharan Desert. Although suspended dust particles are complicated in shape and size that are practically difficult to model in RT simulations, we hypothesize that the dust airmass measured at the site is composed of a mixture of spheres and spheroidal particles for the reasons explained above. The phase matrix associated with the adopted empirically derived aspect ratio distribution was simulated using a modified, substantially faster version of the Light Scattering package for all RT calculations (Lyapustin et al., 2021; Korkin and Lyapustin 2023). See Appendix for a detailed description of how the original Light Scattering package was modified with a specific aim to shorten the computational time without losing accuracy. A test performed to evaluate the differences in the simulation between the original DLS and refactored SDLS software tools revealed minor differences of ~0.001-0.002 in the simulated SSA for the AERONET dust aerosol inversion measurements at the Izana site. Such small differences in the SSA simulations are negligible compared to the expected errors in the spectral SSA due to the uncertainties in the measurement calibration and other assumptions made in the inversion procedure.

## 3.3 MFRSR-AERONET MATCHUP PROCEDURE

The matchup procedure of the AERONET-MFRSR measurements deals with the three kinds of datasets: 1-minute interval measurements of diffuse and direct-normal voltages, AERONET Level 1.5 almucantar combined with hybrid sky scan inversion dataset, and AERONET Level 1.5 direct spectral AOD measurements. While the AERONET Level 1.5 almucantar scans are mostly performed during the early morning-late afternoon hours, ensuring an adequate sky signal at higher solar zenith angles (>50°), hybrid scans are made once per hour throughout the day. Direct measurements of spectral AOD are made typically at 5-minute intervals at the Izaña site. To maximize the usage of the MFRSR's 1-minute hemispheric sky observations, each of these measurements were temporally collocated with the nearest available AERONET inversion instance, either almucantar or hybrid, within a specific day. For instance,








near-noon MFRSR measurements could be matched with either morning or afternoon AERONET inversion data, whichever is closest in time. Similarly, the nearest-in-time direct AERONET AOD measurements were also assigned to each MFRSR data points. This simple temporal collocation procedure provided a matchup dataset which begins with an instance of the AERONET inversion dataset (the real and imaginary part of the refractive index, particle size distribution, and other related parameters) followed by a number of collocated MFRSR measurements that include diffuse and direct-normal voltages and corresponding AERONET spectral AODs. This procedure was performed for each wavelength of the MFRSR independently.

#### 3.4 PROCEDURE FOR AEROSOL ABSORPTION INVERSIONS

The MFRSR-AERONET merged file contains each instance of the AERONET direct and inversion data product values and a corresponding number of collocated MFRSR measurements for a specific day. For the 440-nm wavelength measurements, the real and imaginary parts of the refractive index and the particle size distribution parameters provided by AERONET were used to create input files for the SDLS Light Scattering software package. A total of 21 bins of aspect ratio distribution ranging from about 0.4 (oblate) to about 2.5 (prolate) are prescribed with associated weighting factors shown as the red curve in Figure 5. The 22-bin volume size distribution of AERONET was used as direct input to the SDLS package. Using these parameters as input, the phase matrix elements were simulated at a total of 181 scattering angles at a 1-degree resolution. The resultant phase matrix values along with the total AOD measured by AERONET and trace gas absorption optical depths were then ingested into the ARIZONA RT code to simulate the diffuse and direct irradiance at the prescribed atmospheric pressure. The vertical profile of aerosols is assumed to follow the Gaussian distribution with the peak concentration at 3 km.

We adopted a look-up table-based approach in which the simulation of the ratio of diffuse to direct-normal (DD) irradiance was created using the phase matrix calculated for a total of 15 node values of the imaginary part of the refractive index. The measured diffuse to direct ratio was then fitted linearly to an array of simulations to derive the corresponding imaginary part of the refractive index and SSA. The Jacobians quantifying the sensitivity of the retrieved imaginary part of the refractive index to the simulated DD ratio and the SSA to the imaginary part of the refractive index were also calculated. These Jacobians are useful to estimate error in the retrieved SSA due to uncertainty in the DD ratio measurements. The entire inversion procedure was applied to each of the five wavelengths of the MFRSR independently.

The aerosol absorption inversion procedure described above essentially follows the method described in Krotkov et al. (2005b), except that 1) the aerosol phase matrix for the coarse mode dust aerosols is simulated externally and ingested into the ARIZONA RT model instead of using the Mie theory calculations for spherical particles, and 2) adoption of a LUT-based approach to find the solution in the imaginary part of the refractive index, which is more efficient than an original iterative procedure, which fails for certain observations. A simplified flowchart of the AERONET-MFRSR synergistic algorithm is illustrated in Figure 6.

The entire MFRSR multi-wavelength observations record at IZO was processed with the present algorithm. The inversions were attempted for all observations with  $AOD_{440} > 0.2$  and solar zenith angle  $







number of observations and valid inversions performed for different wavelengths and years. It was noticed that the inversion success rate for the 440-nm wavelength is noticeably lower compared to other shorter wavelengths. The majority of these failed retrieval attempts corresponds to either lower AOD conditions (0.2 < AOD440 < 0.4) and/or larger solar zenith angles (65°-70°), and higher AERONET SSA of ~0.98. Under such conditions and at the minimum node value of 0.0005 in the imaginary part of the refractive index, the measured DD ratios, even after accounting for 1% uncertainty in the measurements, were found to be significantly higher compared to the RT simulations at the minimum node value of 0.0005 in the imaginary part of the refractive index, below which both DLS and SDLS simulations are not recommended. Another source of uncertainty in retrieving aerosol properties at lower AOD conditions at the Izana site is the lower boundary reflectance with bi-directional reflectance distribution function (BRDF) of the mountain slope. Additionally, clouds and the Rayleigh-aerosol radiative interactions from the atmosphere below the station can also modify the spectral sky radiance measured by both MFRSR and AERONET sensors. Accounting for these effects in the inversion algorithm is complicated, which makes retrievals on higher altitude mountain slope stations more complicated, difficult, and uncertain than those at stations with relatively a flat surrounding landscape/terrain.

#### 4 RESULTS

## 4.1 SSA RETRIEVALS AT 440 NM AND ITS COMPARISON AGAINST AERONET

In Figure 7(a), we compare the MFRSR-retrieved SSAs at 440 nm against those of AERONET for a transported Saharan dust event observed between August 26-28, 2020. The SSAs are compared for the coincident AOD ranges 0.2<AOD<sub>440</sub><0.4 (blue circles) and AOD<sub>440</sub>>0.4 (red circles). Vertical gray bars represent 1-standard deviation of MFRSR SSAs retrievals at a 1-minute frequency collocated within  $\pm 30$  minutes of the AERONET inversion. The resulting statistics of the comparison are included within the plot. The remaining three panels of Figure 7 (b, c, d) show the true-color RGB images, visually illustrating the dust transport, captured from the VIIRS sensor onboard Suomi-NPP satellite over the Tenerife Island for the respective days. We find that all matchup data points were contained within the  $\pm 0.03$  error limits, which is the expected uncertainty in the AERONET SSA inversions at AOD<sub>440</sub>=0.4. Sinyuk et al. (2020) have shown that the uncertainty in AERONET SSA decreases significantly for AOD<sub>440</sub> greater than 0.4. Furthermore, the data points (in red) with higher aerosol loading conditions (AOD<sub>440</sub>>0.4) showed a relatively lower SSA in the range ~0.91-0.95 compared to lower AOD conditions (






aerosol loading (AOD~1.0) on August 26, followed by relatively lower aerosol loading (AOD~0.5-0.4) on the following two days. Despite a significant reduction in the total aerosol loading above the Izaña site on August 26 and August 27, the retrieved spectral SSA from the MFRSR shows a similar intraday pattern and magnitude, indicating the stability of the inversion at lower and higher aerosol loadings. The spectrally dependent SSA follows the behavior of the known coarse-mode dust aerosols exhibiting decreasing SSA at shorter wavelengths (Di Biagio et al., 2019). On all three days of dust transport, the SSA at all wavelengths (440 nm to 325 nm) from near-noon to late evening, especially at shorter wavelengths, show an increasing trend. This is also indicative in the AERONET SSA inversions at 440 nm, albeit with a smaller increase relative to the shorter UV wavelengths.

The impact of the choice of the aspect ratio distribution of the spheroidal dust on the retrievals of SSA is also examined. Figure 9 compares the MFRSR-retrieved SSAs at 440 nm under two scenarios: a) using the aspect ratio distribution for randomly oriented spheroidal dust proposed by Dubovik et al. (2006), and (b) employing an empirically derived distribution used in space-based aerosol retrievals over the ocean from near-UV sensors. Compared to AERONET SSA at 440 nm, retrievals based on the Dubovik et al. (2006) distribution show negative biases of 0.021 and 0.012 for AOD<sub>440</sub> ranges of 0.2-0.4 and >0.4, respectively. In contrast, SSA inversions made assuming the empirical distribution yield significantly lower biases of 0.014 and 0.003 for the same AOD ranges. Root-mean-square-difference (RMSD) and number of matchups falling within ±0.03 uncertainty limits also show noticeable improvement. This sensitivity analysis and the corresponding evaluation against AERONET highlight how important it is to make a right choice representing the distribution of dust particle shape in the ground-based dust absorption inversions. Consequently, we adopted the empirical aspect ratio distribution in all on-the-fly RT simulations for creating the LUT required in the inversion algorithm. This choice aligns well with the widely accepted view that transported dust aerosols consist of a mixture of spherical and spheroidal particles. An improvement in the space-based aerosol retrievals from the near-UV sensors further supports this hypothesis (Torres et al., 2018).

A year-by-year comparison of the retrieved SSA at 440 nm between the two sensors is presented in Figure 10. A notable finding is the strong agreement observed in 2020 and 2021, with mean biases of -0.003 and 0.007 and RMSD of 0.013 and 0.011, respectively. In contrast, the MFRSR-retrieved SSA for 2022 and 2023 exhibits relatively larger positive biases of 0.021 and 0.010 and RMSD of 0.024 and 0.017, respectively, in cases with higher aerosol loading (AOD<sub>440</sub> > 0.4). This interannual variability in the apparent bias suggests that the elevated SSA values in 2022 and 2023 may be linked to remaining calibration issues in the MFRSR retrievals. Specifically, the current calibration method for direct and diffuse irradiance is based upon relatively clean conditions (AOD<sub>440</sub> < 0.10) and subsequently applied to days with higher aerosol loading (AOD<sub>440</sub> > 0.2) on a monthly basis. Any changes in the instrument's performance during specific months between cleaner and aerosol-laden conditions, therefore, may introduce systematic errors in SSA retrievals.

## 4.2 MULTI-YEAR UV TO VIS SPECTRAL SSA INVERSIONS

The observations recorded by MFRSR#582 at Izaña have been processed individually at all wavelengths to create a multi-year dataset of spectral aerosol absorption. Figure 11 displays the monthly times series of the retrieved SSAs, in the format of a box and whisker plot, at all five wavelengths. Grey boxes and whiskers represent the 25th-75th






460 percentile range and 1.5 times the interquartile range, respectively; circles and horizonal lines are mean and median of the data for each month. SSA retrievals corresponding to the AERONET AODs at respective wavelengths>0.4 and AERONET Extinction Ångström Exponent< 0.6 (440-870 nm) were included in the timeseries charts. SSA datasets at all five wavelengths, specifically at shorter UV wavelengths, show distinct interannual and intraseasonal variations. First, the SSA at UV wavelengths during the summer months of 2020 and 2023 are found to be lower in magnitudes 465 (more aerosol absorption) as compared to those during the years 2021 and 2022. A large spread in the SSA data was also noticed in the month of July 2020. Second, a well-defined intraseasonal pattern in the SSA can be seen during the summer months, especially in 2021 and 2022, where SSA are found to be lowest in June and gradually increasing until October. An exception to this pattern is the month of August, where the spectral SSA at all wavelengths show a slight dip in the magnitude. Both the interannual and intraseasonal variations in spectral SSA are of prime interest for 470 the estimation of dust radiative forcing, its influence on the atmospheric stability, and aerosol-cloud interactions. The interannual and intraseasonal variations in spectral SSAs, beyond the range of their expected uncertainty of 0.02-0.03, are indicative of diverse sources of dust emissions having distinct absorption properties.

Figures 12, 13, and 14 summarize the combined spectral retrievals from 2019 to 2023 for the imaginary part of the refractive index, the aerosol absorption optical depth, and the single-scattering albedo, respectively. The data are presented separately for the summer months of June through September as box and whisker plots, where orange and light-blue boxes represent retrievals from the MFRSR (440 nm to 325 nm) and AEROENT (440 nm to 1020 nm), respectively. Retrieval instances with AERONET-measured AOD>0.4 and Extinction Ångström Exponent (440-870 nm) < 0.6 are included in the plots to restrict the analysis to moderate to higher dust aerosol loading, thus minimizing the uncertainty in the aerosol absorption inversion at lower AODs in both AERONET and MFRSR datasets. During the peak months of Saharan dust transport to the Atlantic Ocean and over the Izana site (June through September), the imaginary part of the refractive index and AAOD both exhibit a weak spectral trend in the visible to near-IR region (AERONET) but a distinct increasing trend towards shorter UV wavelengths—a typical and expected spectral absorption behavior of coarse-mode dust aerosols (Kandler et al., 2007; Wagner et al., 2012). As result, the SSA shows weak absorption with SSA values in the range 0.98-1.00 at red (675 nm) and longer wavelengths but decreases rapidly at 440 nm with a sharp decline at the UV wavelengths. The aerosol Absorption Ångström Exponent (AAE) values calculated for the MFRSR UV to visible light range (325-440 nm) are found to be noticeably larger than that calculated using the AERONET visible to near-IR dataset (440-870 nm). Furthermore, the AAE values are found to be largest in June (4.52), with a monotonic decrease as the season advances, reaching to the lowest value (2.02) in September. The intraseasonal variations in all three retrieved quantities show the largest absorption, but with significant variability in July and August, followed by relatively moderate absorption in June, and the lowest absorption strength (SSA>0.9) with minimal variability in September.

The spectral imaginary part of the complex refractive index retrieved from both MFRSR and AERONET observations is compared with laboratory-based measurements reported in the literature. Wagner et al. (2012) conducted a detailed retrieval analysis of five different Saharan dust aerosol samples representing diverse mineralogical compositions collected from locations such as Burkina Faso, Cairo, and during the SAMUM field campaign. Using an inversion







scheme based on a spheroidal particle model, the study retrieved refractive indices over the 305–955 nm wavelength range. The retrieved imaginary part of the refractive index (k) exhibited significant variability across the different samples, with k values ranging from approximately 0.005 to 0.02 at ~450 nm, and from ~0.01 to 0.03 at ~350 nm, with overall uncertainties in the range of 0.002 to 0.01. Another Izaña-specific study by Kandler et al. (2007) derived the chemical composition and complex refractive index of transported Saharan mineral dust during July and August 2005, based on a mineralogical model derived from electron microscopy. The study reported a k value of 0.009, which closely aligns with the k value of 0.007 at the same wavelength obtained from the direct optical measurements by Patterson et al. (1977).

Biagio et al (2019) conducted extensive in situ measurements of the imaginary part of the refractive index *k* and SSA over the spectral range 370-950 nm of mineral dust aerosol samples collected from 19 globally distributed dust sources, including the Saharan and Sahelian Deserts. Of particular interest to the present study, the Saharan Desert was represented by soil samples collected from eight locations, where *k* values at 370 nm ranged from 0.0011 to 0.0088, with mean values of 0.0033 and 0.0049 for the North Africa-Saharan and Sahelian samples, respectively. The corresponding in-situ measurements of SSA at 370 nm for those soil samples were found in the range 0.85-0.92. The MFRSR-retrieved *k* values at 380 nm in the present study, on an average, range between 0.003 to 0.006, depending on the month of observations, with an upper bound value of ~0.009 observed in July. The corresponding monthly mean SSA values at 380 nm lie between 0.86-0.88 during June through August. While the averaged dust spectral absorption obtained in Biagio et al (2019) was consistent with that retrieved from AERONET-based inversion (Dubovik et al., 2002) and Balkanski et al. (2007) for dust with a 1.5% volume fraction of hematite content, the spectral K and SSA values across near-UV to near-IR wavelengths were systematically lower than those reported in several earlier in situ and satellite-based studies. Nonetheless, we find that the MFRSR-retrieved K and SSA values at the Izana site align closely with the equivalent in-situ based measurements reported in Biagio et al. (2019) for the North African-Sahara and Sahel desert areas.

The spectral *k* values retrieved from the MFRSR and AERONET in the UV and visible ranges in this study (see Figure 12), and those obtained by Biagio et al. (2019), are significantly lower than those reported in previous studies discussed above. A key factor contributing to this discrepancy, beyond the inherent uncertainties in both in-situ and remote sensing techniques, may lie in the fundamental differences in measurement approaches. Laboratory measurements are typically performed on particles extracted from aerosol samples under controlled conditions, whereas ground-based and satellite remote sensing retrievals represent effective column-averaged aerosol properties in their ambient atmospheric conditions. Additionally, the dust aerosols observed at the Izaña site could be originated from different sources depending on wind directions and speed, potentially leading to observations that reflect either the optical characteristics of a dominant dust source or a mixture of multiple sources.

# 5 CONCLUSION AND FINAL REMARKS

We presented a multi-year, UV-VIS spectral aerosol absorption dataset collected at the Izaña Observatory, Tenerife Island, by the modified ground-based UV-Vis MFRSR instrument. The Izaña site offers a unique setting for ground-based atmospheric dust aerosol remote sensing for two main reasons: 1) its proximity to the Saharan desert enables








sampling of transported dust plumes, and 2) its high elevation of ~2.4 km offers exceptionally cleaner atmospheric conditions on non-dusty days for accurate on-site calibration of the measurements. The MFRSR on-site calibration procedure used in this study, as described in section 2.2, differs from that employed in previous field deployments. Specifically, the partitioning of total spectral irradiance measured by the MFRSR into direct and diffuse components was adjusted, while keeping the total irradiance unchanged, based on space-time collocated AERONET spectral AOD during dusty conditions (AOD440 > 0.2). The modified calibration procedure was applied to the MFRSR measurements on a monthly basis. Furthermore, the aerosol absorption retrieval algorithm was updated from an iterative approach to a LUT-based method. This new method incorporates on-line RT simulations generated for 15 discrete values of the imaginary part of the refractive index. The LUT-based inversion significantly improves stability and resolves convergence issues observed in the earlier iteration-based approach.

The SSA comparison at 440 nm, the common wavelength between AERONET and the MFRSR, showed close agreement, where almost all spacetime collocated matchup data were found to agree within  $\pm 0.03$  uncertainty range of AERONET inversions at AOD<sub>440</sub>>0.4. This agreement was a crucial step toward establishing a consistency between the two independent measurement and inversion techniques. We investigated the impact of two different particle aspect ratio distributions on SSA retrievals: (1) the distribution proposed by Dubovik et al. (2006), which is also used in the AERONET inversion algorithm, and (2) an empirically derived distribution assumed in space-based near-UV aerosol retrieval algorithms. The latter is characterized by a continuous mixture of spherical and spheroidal particles. Our findings showed that using the Dubovik et al (2006) distribution in the inversion tends to underestimate SSA by -0.02 and -0.01 for lower (0.2< AOD<sub>440</sub> <0.4) and higher (AOD<sub>440</sub> > 0.4) aerosol loading conditions. In contrast, the empirically derived distribution resulted in significantly improved agreement with AERONET SSA at 440 nm, exhibiting relatively lower bias. Based on these compelling results, we adopted the empirically derived aspect ratios for all RT calculations used to generate the on-the-fly LUTs employed in the MFRSR inversion algorithm.

The multi-year UV-VIS spectral SSA time series reveal notable intra-seasonal and interannual variability. At shorter UV wavelengths, the retrieved SSA generally exhibits a clear increasing trend from June through October—coinciding with the peak dust transport season—except for August, which consistently shows a deviation with lower SSA values. Interannually, SSA during the summer months increased from 2020 to 2022, followed by a noticeable decline in 2023. Another important finding is the significantly higher variability in MFRSR-retrieved SSA at UV wavelengths compared to the relatively stable and less-variant SSA retrieved by AERONET at visible-near Infrared (440-1020 nm) wavelengths. This pronounced variability in the UV spectral region suggests the presence of dust from diverse sources with different absorption characteristics as detected over the Izaña site, highlighting the distinct sensitivity of UV SSA to varying mineralogy of dust aerosols.

The multiyear UV-VIS spectral aerosol absorption information derived from the MFRSR observations at the Izaña Observatory site in Tenerife Island provides a unique and valuable dataset for studying various aspects of mineral dust aerosols. First, the ground-based aerosol SSA inversions at the UV wavelengths will be crucial for evaluating the








concurrent space-based BUV aerosol absorption retrievals, such as from AURA-OMI, DSCOVR-EPIC, and S5p-TROPOMI. The availability of the ground aerosol SSA inversion datasets eliminates the need to extrapolate satellite-based BUV aerosol SSA to AERONET's 440 nm retrievals, thereby reducing the associated uncertainties in the evaluation of satellite-ground BUV aerosol SSA comparison. Since July 2024, the MFRSR # 582 instrument has been redeployed alongside the ground-level AERONET site at Santa Cruz on Tenerife Island and resumed its operation for collecting the measurements. Deployment of two additional sensors at the AERONET sites in Bozeman, Montana, USA and Yonsei University, Seoul, South Korea is underway. Together, the MFRSR spectral inversion of aerosol absorption from these three sites will contribute to evaluating BUV aerosol SSA retrievals from the recently launched Ocean Color Instrument (OCI) aboard NASA's PACE satellite. The Unified Aerosol Algorithm (UAA) applied to the OCI-measured UV to SWIR spectral observations utilizes the AOD derived from the combined Dark Target and Deep Blue algorithms in constraining the heritage near-UV aerosol algorithm to retrieve SSA and effective aerosol layer height (https://pace.oceansciences.org/data\_table.htm).

Secondly, the spectral aerosol absorption data derived from the MFRSR sensor will be useful to estimate the direct radiative effects of mineral dust aerosols, the surface UV radiation, atmospheric stability, and tropospheric photochemistry. A key result from the MFRSR dust record at the Izaña site is the enhanced dust absorption of solar radiation at UV wavelengths, exhibiting significant temporal variability. These variations are likely linked to changes in the mineralogical composition of dust aerosols. For instance, the iron-oxide content of the atmospheric dust, although in smaller proportion compared to overall mass (~1%-3% by volume, Wagner et al., 2012), can have significant radiative impact in the shortwave spectral domain. Hematite and Goethite are key components of mineral dust determining the strength of light absorption due to their significantly larger (~100 times) imaginary part of the refractive index as compared to other soil mineral components. Among these two components, the imaginary index of hematite is significantly higher than that of goethite at the visible and UV wavelengths (Go et al., 2022, Figure 1). Such pronounced differences in their spectral absorption characteristics enable a clear separation between the two. Biagio et al (2019) also noted that the sample-to-sample variability in the measured k and SSA values of mineral dust samples collected over diverse dust sources were mostly linked linearly to the mass concentration of the iron oxide, hematite, and goethite, and total elemental iron content.

Go et al. (2022) demonstrated the potential of using BUV-visible observations from DSCOVR-EPIC to retrieve hematite and goethite volume fractions and mass concentrations in airborne dust over various arid regions. The spectral aerosol absorption data retrieved from the MFRSR offer similar opportunities of conducting such compositional inference analysis, which is encouraged as a direction for future research. The retrieval of spectral aerosol absorption properties and subsequent inference of the mineralogical composition of ambient dust aerosols will contribute to their accurate representation in Earth System Models, thereby enhancing our ability to quantify their climate impacts.




#### **AUTHOR CONTRIBUTION**

HT and NK conceptualized the study and performed the inversion procedure. JM and GL prepared the merged AERONET-MFRSR dataset, while NK updated the on-site MFRSR calibration procedure and HJ processed the data using the algorithm developed for deriving the inversion dataset presented in this paper. SK refactored and validated the DLS package, referred to here as SDLS for convenience. WG, GJ, SS, DS, KL, and CW, co-authors from Colorado State University and NOAA Central UV Calibration Facility (CUCF), conducted the laboratory calibration of the MSFRSR instrument prior to its deployment at the Izaña site. AB and RG supervised and maintained the daily operations of the instrument at the Izaña Observatory. DF provided the radiative transfer model used for the LUT calculations. HJ led the manuscript writing, with all co-authors contributing to revisions and improvements. OT supervised the overall workflow, ensuring scientific integrity and accuracy.

# CODE AND DATA AVAILABILITY

The code and results shown in the present paper can be obtained from the first authors upon request. The MFRSR inversion dataset presented in this paper will be made deposited in FAIR-aligned data repositories.

# COMPETING INTERESTS

The lead author and at least one of the (co-)authors are members of the editorial board of Atmospheric Measurement Techniques.



#### 620 ACKNOWLEDGMENTS

The authors are grateful to AERONET team and staff for their efforts in maintaining AERONET site at the Izaña site and made the direct sun and inversion products available to the community. The AERONET deployment is a part of the activities of the WMO-Measurement Lead Centre for aerosols and water vapor remote sensing instruments (MLC). The AERONET sun photometers at the Izaña Observatory (IZO) were calibrated through the AEROSPAIN Central Facility (https://aerospain.aemet.es/), supported by the European Community Research Infrastructure Action under the ACTRIS grant (agreement no. 871115).

#### APPENDIX:

# SDLS - Summary of Changes in Light Scattering Simulation (DLS package)

- Here, we summarize the simplifications implemented in the streamlined, swift DLS package (SDLS). DLS (Dubovik et al., 2006) is FORTRAN code that calculates radiative parameters (extinction and absorption coefficients, and the phase matrix) for light scattering by a group of randomly oriented spheroids, based on their optical (real n and imaginary k parts of the refractive index) and geometric (size r and aspect ratio  $\varepsilon$  distributions, D(r) and  $D(\varepsilon)$ , respectively) parameters, and the particle concentration.
- To expedite calculations, DLS utilizes two types of look-up tables (LUTs), also called kernels. First are "main" kernels K of size ~ 5 GB defined on specific grids of all the mentioned parameters. The fixed kernels k are pre-computed from the respective main K-s at a given aspect ratio distribution D(ε). The LUT approach accelerates the calculation of the radiative parameters from the fixed kernels k using, as input, a user-defined (bimodal) size distribution function, particle concentration, and wavelength λ. DLS defines the kernels K and k for a fixed λ = 0.340 μm because the mentioned radiative parameters depend on the size parameter x = 2πr/λ.
  - However, the implementation of the kernel-based approach in the original DLS package was found to be computationally inefficient and time consuming given the 1-minute intervals of a multi-year MFRSR observational record. Both K and K are stored in ASCII (text) format, with arrays of different sizes and shape located in the same file. For instance, the phase matrix as a function of scattering angle is provided for subsequent calculations while the r-grid is stored just for reference. Iterative reading of this data format is time-consuming, and the corresponding reader is not laconic.
  - Next, DLS relies on a computationally expensive cubic spline for interpolating the fixed kernels k from the given r-grid to the user-defined grid  $r_0$ . However, the spline functionality is used inefficiently; its coefficients are calculated inside the loop over  $r_0$ , despite they do not depend on it. This has been noted by other users before us. Furthermore, DLS calls the spline over r for two points of the n-grid interval that contains the user's value  $n_0$  (same for k-4 calls total). Then, bilinear interpolation is used to get the output at the user's  $n_0$  and  $k_0$ . By switching the sequence of the interpolations first bilinear over the refractive index, then spline over the size the number of spline calls is reduced from 4 to 1. Even that single spline call can be eliminated since we integrate over the size, making interpolation





between size grids unnecessary. Note that instead of the lognormal size distribution  $D(\ln(r))$ , it is more natural to use 655 the size parameter distribution  $D(\ln(x))$  for integration over particle size.

To overcome the noted inefficiencies, we have implemented the following changes to the DLS package for deriving an equivalent but much faster and efficient SDLS software tool.

- The original FORTRAN code was translated to C/C++, as this work was initiated as part of translation of MAIAC's (Lyapustin et al., 2021) polarized radiative transfer solver IPOL (Korkin and Lyapustin, 2023) from FORTRAN into C.
- 2. We now store k in binary files; auxiliary parameters like grids of scattering angles and particle size are hardcoded. As a result, reading k requires just a single command in C.
- 3. All splines were dropped from the code, and we adopted integration in the ln(x) space using the same x-grid as in the fixed kernel k; the same applies to the scattering angle we use the grid from the fixed kernels, 0° to 180° in steps of 1°.
- 4. The DLS code is split into two parts with refactoring applied to only the part utilizing the fixed kernels *k* because it is being called iteratively in retrieval algorithms. Hence, its high efficiency is crucial, especially for the hyperspectral measurements. We have not changed the part of DLS that calculates *k* from *K*; nor have we updated the "main" LUTs. However, we note that keeping the component calculating radiative single scattering properties (called in real time) and the one calculating *k* from *K* within a single code is impractical.

In summary, SDLS software performs 3 steps: (a) reads the fixed kernels k from 7 binary files (six for the elements of the phase matrix, one for the scattering and extinction coefficients combined); (b) performs bilinear interpolation over the real and imaginary parts of the refractive index for the user-provided values; (c) integrates over the size parameter. The refactored package has been tested and is now available from <a href="https://github.com/korkins/spheroids">https://github.com/korkins/spheroids</a> with documentation.

Our improvements are relevant to the software. We added nothing to the scientific aspects described in Dubovik et al. (2006). We are aware that the GRASP Team (<a href="https://www.grasp-open.com/">https://www.grasp-open.com/</a>) has also improved the DLS package. However, we have not evaluated their implementation yet. Instead, we worked with a version provided to us by the original developers, which has served our in-house community, with occasional updates, for about 2 decades and tailored that stable version of DLS package to our specific needs.

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

Table 1 MFRSR-AERONET, satellite sensors and their respective datasets used in the present study.

| Platform/              | Operation Period         | Measurement Characteristics                             |  |  |  |  |
|------------------------|--------------------------|---------------------------------------------------------|--|--|--|--|
| Instrument             |                          |                                                         |  |  |  |  |
| MFRSR Instrument # 582 | July 2019 - January 2024 | Total and Diffuse Irradiance at 305, 311, 325, 332      |  |  |  |  |
|                        |                          | 340, 380, 440 nm                                        |  |  |  |  |
|                        |                          | Temporal frequency: 1 minute                            |  |  |  |  |
| AERONET                | July 2019 – January 2024 | Direct Measurements: Version 3 Level 1.5                |  |  |  |  |
| AERONEI                | Contemporary to MFRSR    | Direct Measurements. Version 3 Level 1.3                |  |  |  |  |
|                        | Contemporary to WEKSK    | Spectral A ODs at 225, 222, 240, 280, and 440 and       |  |  |  |  |
|                        |                          | Spectral AODs at 325, 332, 340, 380, and 440 nm         |  |  |  |  |
|                        |                          | Extrapolated AODs at 325 and 332 nm                     |  |  |  |  |
|                        |                          | Inversion Dataset: Version 3 Level 1.5                  |  |  |  |  |
|                        |                          | Particle size distribution, Real part of the refractive |  |  |  |  |
|                        |                          | index at 440 nm.                                        |  |  |  |  |
| Aura/OMI               | July 2019 - January 2024 | Total column O <sub>3</sub> and NO <sub>2</sub> amounts |  |  |  |  |
| SNPP/OMPS              | July - September 2023    | Total column O3 amount                                  |  |  |  |  |
| Pandora                | July - September 2023    | Total column O <sub>3</sub> and NO <sub>2</sub> amounts |  |  |  |  |

**Table 2** MFRSR data availability at Izaña observatory at different wavelengths and corresponding valid numbers of aerosol absorption inversions. The multiplying factors for each MFRSR wavelength to create nodes in the imaginary part of the refractive index are listed at the end of the table.

| Wavelength [nm] | 2019-08           |                  | 2020              |                  | 2021              |                  | 2022              |                  | 2023              |                  |
|-----------------|-------------------|------------------|-------------------|------------------|-------------------|------------------|-------------------|------------------|-------------------|------------------|
|                 | N <sub>meas</sub> | N <sub>Inv</sub> |
| 440             | 1531              | 764<br>(50%)     | 10849             | 8297<br>(76%)    | 13245             | 7853<br>(59%)    | 9431              | 5968<br>(63%)    | 9192              | 5460<br>(59%)    |
| 380             | 1707              | 1119<br>(66%)    | 11019             | 9525<br>(86%)    | 14182             | 10856<br>(77%)   | 9586              | 8317<br>(87%)    | 9904              | 7939<br>(80%)    |
| 340             | 1739              | 1306<br>(75%)    | 11102             | 9741<br>(88%)    | 14623             | 11578<br>(79%)   | 9878              | 8826<br>(89%)    | 10193             | 8313<br>(82%)    |
| 332             | 1740              | 1322<br>(76%)    | 11111             | 9759<br>(88%)    | 14643             | 11646<br>(80%)   | 9912              | 8851<br>(89%)    | 10244             | 8425<br>(82%)    |
| 325             | 1741              | 1275<br>(73%)    | 1117              | 9704<br>(87%)    | 14663             | 11558<br>(79%)   | 9932              | 8827<br>(89%)    | 10285             | 8494<br>(83%)    |

 $N_{meas}$  = Total number of MFRSR measurements with AOD>0.2 and SZA

Figure 1. (Left) A synoptic view of the northwestern Sahara, Tenerife Island, and the location of the Izaña observatory seen in the Aqua/MODIS true-color RGB image acquired on June 3, 2019. (Right) An aerial photograph showing the installation of MFRSR instrument # 582 and AERONET CIMEL sun photometer at the Izaña observatory.

Figure 2. Normalized spectral response functions of the MFRSR head # 582 wavelength bands calculated prior to its deployment at the Izaña Observatory. Central Wavelength (CW), wavelength range (λ<sub>min</sub>-λ<sub>max</sub>), and full-half-maximum-width (FWHM) values for each band are printed within the plot.

Figure 3. (Left) Examples of MFRSR Step 1 calibration plots on the cleaner days of August 20 and 31 of 2020, when AERONET-measured AODs (440 nm) were quite lower (

Figure 4. An example of a) step-2 and b) step-3 of the calibration procedure for a dusty day of August 26, 2020. a) A

870 constant value of ln(V<sub>0</sub>) calculated from the step-1 is applied to the entire day of observations to calculate the corrected direct-normal voltage, shown as dark-blue dots on (b). The AERONET and MFRSR measured AODs are shown in orange and purple, respectively, on the right-hand side y-axis The corrected diffuse voltage (red curve), shown in (b), is calculated by subtracting corrected direct horizontal voltage from the cosine-corrected measured total voltage. c) the true-color RGB image acquired on the same day from Aqua-MODIS, showing the transport of massive dust plume over the observation site in Tenerife Island and adjacent Atlantic Ocean.

**Figure 5.** Aspect Ratio distribution distributions and corresponding weighting factors representing randomly oriented spheroids. The black and red curves represent the aspect ratio distribution data obtained from Dubovik et al. (2006) and the empirically derived distribution used in the present study, as explained in the text.

**Figure 6.** A simplified flowchart of the MFRSR-AERONET synergistic algorithm for the inversion of the UV-VIS spectral imaginary part of the refractive index and SSA.

**Figure 7.** a) Comparison of the MFRSR vs. AERONET retrieved SSA at 440 nm for the data acquired on August 26, 27, and 28 of 2020. Blue and red circles represent temporally matched data with coincident AERONET-measured AOD (440 nm) 0.2-0.4 and >0.4, respectively. The statistics of the comparison for different AOD conditions are included within the plots. The true-color RGB images acquired from Suomi-NPP/VIIRS sensor for three dates are shown in b), c), and d).

890

**Figure 8.** Sub-hourly spectral SSA retrieved from UV-VIS MFRSR observations acquired on August 26, 27, and 28 (a, b, c, respectively) of 2020. Each circle represents 30-minute averaged SSA values with corresponding averaged AOD printed next to it. Vertical lines represent the expected uncertainty in the retrieved SSA, which combinedly accounts for both a 1% error in the diffuse to direct irradiance ratio and the standard deviation of 1-minute retrievals within a 30-minute time window. Daily mean values of SSAs and AODs are printed on the bottom of each plot.

900 Figure 9. Comparison of MFRSR (y-axis) vs. AERONET (x-axis) SSA at 440 nm for the matchup data collected during 2020 at Izaña site derived assuming a) original aspect ratio distribution representing randomly oriented spheroidal dust suggested in Dubovik et al. (2006) and b) empirically derived aspect ratios adopted in the over-ocean dust aerosol models used for near-UV aerosol algorithm of OMI, EPIC, and TROPOMI sensors. SSA matchup data points in blue and red represent the coincident AOD<sub>440</sub> conditions of 0.2-0.4 and >0.4, respectively. The statistics of 905 the comparison for different AOD conditions are included within the plots.

**Figure 10.** Comparison of MFRSR (y-axis) vs. AERONET (x-axis) SSA at 440 nm for the 2020-2023 (a though d) operation period of MFRSR measurements at Izaña site derived assuming empirically derived aspect ratios. Matchup data points in red (blue) represent  $AOD_{440}>0.4$  (0.2

Figure 11. Monthly time-series charts of MFRSR-retrieved spectral SSAs. The data are represented as box and whisker plot; grey boxes and whiskers represent 25th – 75th percentile range and 1.5 times the interquartile range, respectively. Data with AERONET AOD at respective wavelengths>0.4 and Extinction Ångström Exponent

Figure 12. Spectral plots of the imaginary part of the refractive index retrieved from the MFRSR (325-440 nm) and AERONET (440-1020 nm) instruments. The spectral SSA dataset represents AE<0.6 and AOD<sub>440</sub>>0.4 conditions and are shown as a box-whisker plot for each wavelength separately for the months of June, July, August, and September 2019-2023 in top to bottom order.

Figure 13. As in Figure 12 but for the retrieved spectral aerosol absorption optical depth (AAOD). The aerosol

Absorption Ångström Exponent (AAE) values for UV and VIS-NIR wavelength ranges are also included in each

AAOD plot.

Figure 14. Same as in Figure 12 but for the retrieved spectral SSA.