# Peer review of "Ground-based MFRSR UV-Vis spectral retrievals of Saharan dust absorption at Izaña Observatory"

_EGUsphere, 2025_

## Author Comment (AC1)

**Ground-based MFRSR UV-Vis spectral retrievals of Saharan dust absorption at Izaña Observatory**

Hiren Jethva, Nick Krotkov, Omar Torres, Jungbin Mok, Gordon Labow, Elena Lind, Tom Eck, Wei Gao, George Janson, Scott Simpson, Darrin Sharp, Kathy Lantz, Charles Wilson, Africa Barreto, Rosa García, Sergey Korkin, David Flittner

**Response to Referee # 1**

*\*Reviewer's comments: In blue-italic fonts*

**\*Authors' comments: In black-regular fonts**

Thanks for taking time to review our manuscript and providing constructive and useful comments. Below is the one-to-one response of each comment we hope you find satisfactory. Your comments have helped us in clarifying important aspects of the synergy algorithm and improving the scientific value of the paper.

**Major comments**

*Section 3.2:*

*I found the whole approach of particle shape justification rather questionable.*

*First of all, authors suggest different shape distribution, that is quite badly justified, instead of one that was explicitly designed for the observations they use in their synergetic retrievals. Satellites and ground-based observations work in different scattering angle ranges.*

*Dubovik et al., 2006 and AERONET uses spheroidal model ONLY in a combination with spheres, where spheroidal particles represent an extreme case of non-spherical particles, and the fraction of spheres is the parameter that is fitted from sky observations and it is never 0. Why not use that one observed by AERONET, instead of basically turning shape distribution inside out and creating inconsistency between different parts of synergy?*

Following reviewer's suggestion, we reproduced the AERONET inversion approach, in which irradiance simulations generated for spheroidal aspect ratio distributions (Dubovik et al., 2006) and for spheres are linearly combined, with the sphericity fraction used as a weighting parameter. The resulting MFRSR SSA retrievals obtained using this "mixing" approach exhibited even poorer agreement with AERONET SSA at 440 nm. Please refer to our response to this comment given later in this report for additional details.

*Lines 366-370: "A total of 21 bins of aspect ratio distribution ranging from about 0.4 (oblate) to about 2.5 (prolate) are prescribed with associated weighting factors shown as the red curve in Figure 5. The 22-bin*

*volume size distribution of AERONET was used as direct input to the SDLS package. Using these parameters as input, the phase matrix elements were simulated at a total of 181 scattering angles at a 1-degree resolution."*

*Did authors compared at 440 nm with AERONET provided phase matrices? These will include a "proper" mix of spheres and spheroids that fits the almucantar observations. Also, it would be a nice exercise to see the AERONET provided phase functions for several cases and ones calculated using same refractive index, PSD but using the suggested shape distribution. Ideally simulate almucantar observations to compare the fits. If fitting difference are not that substantial, it also can provide additional justification of the proposed method. Possibly fits of the shorter WL will be even better with updated shape distribution.*

Ingesting AERONET-derived phase functions into the MFRSR inversion algorithm would require a substantial overhaul of the existing inversion package. However, we've compared the AERONET-retrieved phase function F11 at 440 nm with that derived from our proposed empirical one for a single collocated measurement acquired on August 26, 2020, and included in the revised Figure 5. While both scattering phase function F11 show a close correspondence for scattering angle < 140°, they show some differences at larger angles where F11 from empirical aspect ratio distribution show larger values than that of AERONET. The corresponding SSA values, however, were in close agreement.

Regarding adoption of the AERONET inversion approach, please refer to our response to this comment given later in this report for additional details.

*In general, it would be also nice to see validations of AODs estimated using the retrieved absorption and assumed size/shape distribution and real refractive index in UV with AERONET observations (e.g. 340 and 380), I believe Izana should had several CIMELs capable of providing such data.*

In the present synergy inversion algorithm, AERONET spectral AODs are used to derive the calibration constant, $ln(V_0)$. The MFRSR spectral AODs, computed from the direct-normal radiance obtained in this step, show excellent agreement with the AERONET AODs, with differences negligible at the fourth to fifth decimal place. However, the concern raised reflects a circularity in the methodology: aerosol absorption at UV wavelengths is retrieved by constraining the inversion with AERONET-measured AOD and particle size distribution. Although a different particle shape factor distribution is assumed, the inversion remains mathematically constrained by the initially prescribed AOD. Consequently, recomputing extinction AOD from the retrieved parameters necessarily reproduces the same AOD that was assumed in the first place.

*Also, I got completely lost how LUT's are generated/used. Are they dynamic and depend on AERONET due to the multiplication factors? Are they static and calculated to a specific grid? Please, provide more details.*

The LUTs are generated dynamically for each 1-minute MFRSR measurement. The LUT nodes for the imaginary part of the refractive index are constructed by multiplying the collocated 440-nm AERONET imaginary refractive index by a set of predefined scaling or multiplying factors listed in Table 2. This approach yields an array of imaginary refractive indices used in RT simulations of the diffuse-to-direct irradiance ratio. The observed diffuse-to-direct ratio is then fitted into the simulated array to retrieve the k and SSA via linear interpolation. These steps have been clarified in the revised text.

**Minor comments**

*Line 246: "inversion parameters of PSD and the real part of the refractive index" it is not clear how the real refractive index is extrapolated to UV, please clarify*

The real part of the refractive index was not extrapolated to UV wavelength. Due to the lack of real-time quantitative information in the UV region, we assigned the AEROENT 440-nm real part of the refractive index to all shorter UV wavelengths (325-380 nm).

*Line 373: "The vertical profile of aerosols is assumed to follow the Gaussian distribution with the peak concentration at 3 km." Any particular reason to use this profile? And 3km is above sea level or Izana station? And what half width was assumed?*

CALIOP vertical backscatter measurements of the transported Saharan dust aerosols over the Atlantic Ocean and at the Izana site show that dust aerosol layers are generally confined between 1-6 km with peak concentration around 3-4 km above sea-level. Although the shape of the backscatter vertical profile may not exactly mimic that of a Gaussian distribution, the latter is a closer approximation to the real-world aerosol profiles used by satellite retrievals for elevated aerosols. The assumed 3-km aerosol layer height is with reference to above the ground-level, i.e., for the Izana site, which is located at an altitude of ~2.4 km, the aerosol layer is ~5.4 km above sea-level. The Gaussian distribution assumes halfwidth of 0.5 km.

To further investigate the effect of aerosol layer height on the retrieved SSA, we conducted an inversion run for MFRSR observations acquired on August 26-28, 2020—the three case studies shown in Figure 10 (revised paper), by perturbing the assumed aerosol height by ±1 km, i.e., 2 km and 4 km. The resultant retrievals of SSA were compared against those retrieved assuming 3 km aerosol height. For the higher AOD case on 26 August (AOD ≈ 1), the resulting changes in retrieved SSA were approximately ±0.0002, ±0.005, and ±0.007 at 440, 380, and 340-325 nm, respectively. For the lower AOD cases on 27-28 August, SSA errors at these wavelengths were relatively much smaller (<±0.0005). Increasing (decreasing) aerosol layer height is found to produce positive (negative) errors in SSA. Overall, the errors in the retrieved SSA induced by uncertain aerosol layer height are minor compared with those arising from individual uncertainties in the input AERONET AOD and MFRSR measurements.

*Line 380: "The entire inversion procedure was applied to each of the five wavelengths of the MFRSR independently."*

*Was spectral dependence controlled in any way? Are there any examples how spectral behaviour of such retrievals looks like? Is it reasonable? Has it spikes, does it have a trend? Would be nice to see plots of examples of full spectrum imaginary refractive index, combined with AERONET data just to have a glimpse what could be expected from dust using this technique.*

No, the spectral dependence was not controlled in any way in our synergy algorithm. The retrievals of aerosol absorption were performed at each MFRSR wavelength independently. The spectral behavior of the combined UV-VIS spectral imaginary part of the refractive index ($k$), SSA, and AAOD, along with corresponding AERONET spectral values (440-1020 nm) are already presented in **Section 5.2** and shown in **Figures 14, 15, & 16** in the revised paper. All three absorption quantities ($k$, SSA, AAOD) in the UV spectral region follow our expectation with increased absorption at shorter wavelengths, which is in-line

with several in-situ measurements published in the literature. Noticeably, the Absorption Ångström Exponent (AAE) calculated from spectral AAOD in the UV region is found be higher and in the range 1.0-4.6 (depending on the month of observations) against that in the range 1.6-2.9 derived from AERONET 440-1020 nm wavelength range.

*Line 390: what retrieval is considered a "success"? Please, clarify. Are they treated case-wise or wavelength-wise, for e.g.? If one channel "failed" is all retrieval discarded?*

"Successful" retrievals are defined as those in which both *k* and SSA are derived for each 1-minute MFRSR observation at a given wavelength independently. If a retrieval fails at one wavelength but succeeds at others, the successful results are NOT discarded but retained and reported for the corresponding wavelengths. Consequently, the number of successful retrievals may vary across wavelengths. This clarification has been added to the revised manuscript.

*Lines 433 – 439, Line 550: I'm not sure such comparison is rather fair. First of all as mentioned above there should be certain percentage of spherical particles retrieved by AERONET, so the distribution won't be exactly as in Dubovik 2006, and maybe be somewhat closer in resulting phase function to what authors suggest. Also it is not clear do they compare sucessed cases of their retrievals only or all of them, maybe choise of shape affect success rates? Also I'm confused how method using the same refractive index, same psd and as claimed same shape distribution as in AERONET (case a) shows bias with AERONET retrieval itself, I mean these SSA values retrieved under exactly the same assumptions, it is clear that the shape distribution can't be the not only reason in that case.*

We've compared the scattering phase matrix F11 at 440 nm simulated for a single set of AERONET-MFRSR collocated measurement for the three representative dust case studies demonstrated in Figures 9 and 10 of the revised paper. The figure shown below (Figure 5 in the revised paper) illustrates the F11 comparison of mineral dust aerosols retrieved on a) August 26 at 10:10 UTC, b) August 27 at 10:19 UTC, and c) August 28 at 15:09 UTC of 2020 corresponding to randomly oriented spheroids proposed in Dubovik et al. (2006) (black), empirically derived distribution (red) used in the present study, and that from AERONET inversion product (blue). F11 (except for AERONET) was simulated using the SDLS software package for the respective dates. Aspect ratio distribution proposed by Dubovik et al. (2006) and that used in the presented study are shown as an inset in the top-left plot (a). While the F11s closely correspond to each other for scattering angles < 120°, F11 associated with the empirical (fixed) aspect ratio distribution deviates noticeably for angles > 120°. Despite these differences, the corresponding retrieved SSAs (black and red) show a close agreement—indicating that the choice of aspect ratio distribution didn't play a significant role in the inversion at least for these three dust events.

Additionally, we also tested the AERONET-like "*mixing*" approach in the synergy algorithm, in which the algorithm was applied separately for each 1-minute MFRSR observation assuming 1) Dubovik et al. (2006) aspect ratios of randomly oriented spheroids and 2) spheres separately. The simulated diffuse-to-direct irradiance ratios were then linearly mixed by using the "sphericity" parameters of AERONET as follows:

$$DDRatio_{LUT} = DDRatio_{sphere} * Sph_{frac} + DDRatio_{spheroiod} * (1 - Sph_{frac})$$

Where, *DDRatio_{LUT}* is the sphericity-weighted simulation of diffuse-to-direct (DD) ratio simulation (LUT), *DDRatio_{sphere}* is the simulation for purely spherical particles, and *DDRatio_{spheroid}* is the simulated DD ratio simulated using the Dubovik et al. (2006) aspect ratio, and *Sph_{frac}* is the AERONET-provided sphericity fraction.

[Figure]

Figure 1 Scattering phase function F11 (440 nm) of mineral dust aerosols retrieved on a) August 26 at 10:10 UTC, b) August 27 at 10:19 UTC, and c) August 28 at 15:09 UTC of 2020 corresponding to randomly oriented spheroids proposed in Dubovik et al. (2006) (black), empirically derived distribution (red) used in the present study, and that from AERONET inversion product (blue). F11 (except retrieved from AERONET) was simulated using the SDLS software package for a single MFRSR-AERONET collocated measurement for the respective dates. Aspect ratio distribution proposed by Dubovik et al. (2006) and that used in the presented study are shown as an inset in the top-left plot (a).

The *DDRatio_{LUT}* as a function of the nodes in the imaginary part of the refractive index are then used to retrieve *k* and SSA. This new version of the algorithm was applied to the year 2020 MFRSR observations at 440 nm. Figure (a) shown below displays the comparison of the retrieved SSA against that of AERONET. We find that the negative bias in the MFRSR-retrieved SSA (440 nm) is further increased, compared to spheroid-only results, to ~-0.02, whereas the comparison shown in (b) derived using the empirical aspect ratios yields improved comparison with mean bias of just -0.003.

[Figure]

Figure 2 Comparison of MFRSR (y-axis) vs. AERONET (x-axis) SSA at 440 nm for the matchup data collected during 2020 at Izaña site derived assuming a) AERONET-like 'mixing" approach combining sphericity-weighted randomly oriented spheroid (Dubovik et al., 2006) and sphere and b) empirically derived aspect ratios adopted in the over-ocean dust aerosol models used for near-UV aerosol algorithm of OMI, EPIC, and TROPOMI sensors. SSA matchup data points in blue and red represent the coincident AOD$_{440}$ conditions of 0.2-0.4 and >0.4, respectively. The statistics of the comparison for different AOD conditions are included within the plots.

We agree with the reviewer that the aspect ratios may not be the only factor attributing to the negative bias in SSA when the retrievals are performed under the same assumptions, except that the real part of the refractive index, which is assumed to be the same at UV wavelengths as at 440 nm.

The choice of empirical aspect ratios in the present MFRSR aerosol absorption inversion is made because it provided a better comparison against those of AERONET at 440 nm with significantly reduced bias.

*Figure 12-13: Why wishers are so much bigger for July-August? Please discuss*

The figure numbers are changed to 14-15 in the revision. Bigger whiskers represent significant interannual variability of the retrieved absorption quantities, indicating varying mineral composition of dust originated from different areas of Sahara. The spread in the retrievals is also reflected in the timeseries charts (Figure 13 in the revision), especially in the month of July and October. While a separate analysis is required, which is out-of-scope for the present study, to pinpoint exact sources of these aerosols, it is discussed in the first and last paragraphs of Section 5.2 as well as in conclusion section.

*Line 482-485: "the imaginary part of the refractive index and AAOD both exhibit a weak spectral trend in the visible to near-IR region (AERONET) but a distinct increasing trend towards shorter UV wavelengths— a typical and expected spectral absorption behavior of coarse-mode dust aerosols" if I understood correctly "multipication factors" in table 2 there's little to no chance that method will retrieve imaginary part of refractive index below the one of AERONET, and it seems that a trend for decreasing absorption with wavelengths is kinda "bult-in" through these factors.*

The Reviewer is not quite right here. The multiplication factors for all wavelengths go much beyond 1.0 and up to 0.4 (440 nm) and 0.5 (for all UV wavelengths), as included in Table 2. For example, for a given AERONET imaginary part of the refractive index of 0.001, the corresponding nodes generated in the inversion code would be array of [0.0004 0.0006, 0.0008, 0.0010, 0.0012, 0.0014, 0.0016, 0.0018, 0.0020, 0.0030, 0.0040, 0.0050, 0.0060, 0.0070, 0.0080]. So, the retrieved *SSA* can go above the AERONET reported value at 440 nm if the MFRSR DD ratio observation fits into the LUT.

*Table 2: It is not clear how "multiplying factor" are used actually these are important and not mentioned anywhere else. It is a significant flaw in method description. Also if imaginary part of refractive index is retrieved a a factor to AERONET it is not completely clear how LUTs are generated, are they individual for every case? Or it is the factors that are retrieved, please, provide a more comprehensive description of this part of the method. And why such specific selection of factors? They are quite different for the UV and blue for e.g.*

The nodes in the imaginary part of the refractive index are generated dynamically for each MFRSR observations. The collocated *k* value of AERONET at 440 nm is multiplied by the factors, included in Table 2, to generate an array of *k* values. The RT model is run on these *k* values, along with AOD, phase matrices and other input parameters, to generate one-dimensional array of simulated diffuse-to-direct ratios or LUT. The MFRSR-measured DD ratio is then fit linearly into the LUT to retrieve *k* and SSA. The description is further clarified in the paper.

*Line 657: "The original FORTRAN code was translated to C/C++, as this work was initiated as part of translation of MAIAC's (Lyapustin et al., 2021) polarized radiative transfer solver IPOL (Korkin and Lyapustin, 2023) from FORTRAN into C."*

*It is not clear which translation is mentioned, was code manually re-written in C? FORTRAN and C share compiler and their translator makes same pseudocode for further compilation, this doesn't affect the speed of execution.*

*Generally the whole Appendix part of the DLS package modifications looks a bit weird to me. Especially for a user of DLS package. It looks like the package wasn't used in the optimal way, and instead of changing several parameters in the it was re-written… I presume the explicit permissions for such code use were provided.*

*Majority of statements are either not directly related to the DLS package performance, but rather to the use case that was not optimal, FORTRAN and C binds naturally so the whole C translation for the performance looks a bit superficial.*

A separate response file is uploaded with this response that addresses reviewer's comment on SDSL software tool.

*Besides authors keep saying that LUTs containing imag parts were used for the retrievals, i.e. multiple running and reading of phase functions kernels as well as RT calculations for different imaginary parts*

*supposedly was done only once, and then LUTs were re-used, or I'm missing something, please provide more details on this.*

We find this as a useful suggestion. In the current framework, multiple 1-minute MFRSR observations are associated with a single, collocated AERONET inversion. In principle, an on-the-fly LUT—constructed over a fixed grid of the imaginary part of the refractive index—could be generated once and then reused for all subsequent MFRSR observations linked to that AERONET inversion. This procedure could be repeated for each AERONET inversion and its corresponding set of collocated MFRSR measurements. However, the runtime of original DLS software (i.e., the number of AERONET inversions x 10 nodes in the imaginary index) still remains longer than that of SDLS software tool. Moreover, this change will require restructuring of the inversion code demanding time and efforts. Given very fast processing time/speed of the SDLS package, our inversion code executes SDLS package to create LUT for every 1-minute measurements. To further reduce computational cost, the inversion code is designed to terminate simulations once the MFRSR observation falls between two adjacent simulation nodes.

**Technical comments**
*Line 117: "multiple" I'd suggest replacing with "five"*
Suggestion accepted.

*Line 121: "these wavelengths", are these 6 or 5?*
The sentence has been revised as "to derive the imaginary part of the refractive index independently at 440 nm and UV wavelengths, except 311 nm".

*Line 392: "higher AERONET SSA", please provide wavelength, is it 440?*
Yes, the SSA here corresponds to 440 nm. Added in the text.

*Figure 8: Consider making it double Y plot with AOD on the right, it is bit messy, too many fine text in color around points, quite hard to analyse.*
Adding Y plot on the righthand side axis will add more data points (circles) inside the plot, which will make plot further crowdy. We prefer to maintain the current format of this plot (Figure 10 in the revised text).

*Line 431: "440 nm to 325 nm" I would suggest "325 to 440" this way it will be clearer where trend increases.*
Changed to "325 nm to 440 nm".

*Figure 11: Consider making text bigger, and what are these tiny numbers below?*
Font size is increased. The numbers printed below the whiskers are sampling for each month.

*Figure 12-13: Generally hard to follow spectral and temporal dependencies and the font is rather small and hard to read, is there a better way to present these data?*
The paper now includes revised Figures 14, 15, and 16 formatted with bigger fonts, contrasting colors, and increased horizontal dimension (size). Also, the order of the figures 15 and 16 is reversed with SSA results shown first followed by AAOD as it makes more sense.

**Response to Reviewer 1 on DLS/SDLS Tool**

An explicit permission to modify and distribute the spheroidal package, originally developed and reported in Dubovik et al., JGR, 2006 (https://doi.org/10.1029/2005JD006619) seems to be the main Reviewer's concern regarding the DLS software. We address this comment first and turn to other comments – also important but technical – after that. Not necessarily in the order the issues were mentioned, but each one was addressed. The Reviewer's statements are quoted in "*Blue italic*"; our modifications to the section are in **Bold** (both original and revised sentences are given, marked as [original] and [revised], respectively).

"*I presume the explicit permissions for such code use were provided.*"

On October 23rd, 2023 an email exchange between Coauthor Dr. S. Korkin and the lead developer of the DLS package Dr. O. Dubovik took place. As an outcome of that email exchange, it was agreed that every time Dr. Korkin uses the modified SDLS, he will mention that:

a) GRASP team has their own, stand-alone, improved modification of the legacy package: https://code.grasp-open.com/open/spheroid-package/ (simple registration is required).
b) The refactored Dr. Korkin's version originated from a decade-old version of the legacy DLS code which Dr. Korkin obtained in 2011 from its developers.
c) All Dr. Korkin's changes are limited to the reader and interpolation code and no changes in science (e.g., values in the kernels, definition of the particle size grid, etc.) have been made.

The last paragraph of the SDLS Appendix fulfills the agreement. Dr. Dubovik wrapped up the abovementioned email exchange with this phrase (exact quotation) "*In brief, we have fulll understanding . No problem.*"

"*I'm no expert in this, but I believe a clear statement that original DLS package re-use was done with explicit permission of its authors is required in this appendix*"

We believe our effort in refactoring the original DLS package is fair because:

a) Original author agreement to user modifications - see (a, b, c) above.
b) We worked with a decade-old package, not the recent one, therefore not throwing shade on modern GRASP ( https://www.grasp-earth.com/grasp-open/ ). Note, however, that the legacy version is still used widely.
c) The paper about the package, Dubovik et al., JGR: 2006, says nothing about license. Instead, it says (page 8, before Sec.3): "*The kernels and software package with a detailed description of its functions is publicly available from the lead author upon request.*"
d) If no specific license is provided, we assumed GNU General Public License (GPL, which has been in place since 2007 – thus covering the moment we obtained the package). Quoted from their website ( https://www.gnu.org/licenses/gpl-3.0.en.html ): "*GNU General Public License is intended to guarantee your freedom to share and change all versions of a program--to make sure it remains free software for all its users.*" (in Preamble)

e) We make no profit from the refactored version and distribute SDLS with no specific license attached (the Reviewer has noted the link to the GitHub repository).

"... *or proper authorship affiliations should be provided in the linked repository. Otherwise it gives a rather weird feeling to say the least. For e.g. git repository contains binary kernels that contain transformed information from the text files of the original DLS package without any authorship affiliation nor licence mentioned,* [… later we respond to this missing part questioning, as we understand, about how SDLS was tested …] *To be frank, these* [the kernels] *are the essence of the package, non-spherical part being the important improvement in this study, and making these from scratch is not as easy as loading and interpolating between the already calculated nodes. And the only "link" with the kernels authors in repository with its authors is an image, representing a screen shot of the original article in the doc section...*"

Yes, we missed that: at the moment of submission, our GitHub Readme was showing "DLS refactoring". The SDSL Appendix from the manuscript, with full title and link to the original paper, is now added to the repository.

We have also amended the first sentence in the SDLS Appendix as shown below in order to say, at the very beginning, that only the loading/interpolating part was altered:

[original] "Here, we summarize the simplifications implemented in the streamlined, swift DLS package (SDLS)."

[revised] "Here, we summarize the simplifications implemented in the streamlined, swift DLS **reader/interpolator** (SDLS)."

The sentence before the numbered list and the first bullet are changed as follows:

[original] "To overcome the noted inefficiencies, we have implemented the following changes to the DLS package for deriving an equivalent but much faster and efficient SDLS software tool.

1. The original FORTRAN code was translated to C/C++, as this work was initiated as part of translation of MAIAC's (Lyapustin et al., 2021) polarized radiative transfer solver IPOL (Korkin and Lyapustin, 2023) from FORTRAN into C."

[revised] "To overcome the noted inefficiencies, we have implemented the following changes to the DLS package for deriving an equivalent but much faster and efficient **SDLS tool for reading and interpolating the fixed kernels generated by the original DLS package**.

1. **A few subroutines of the original DLS FORTRAN code responsible for reading and interpolating the fixed kernels were manually** translated to C/C++, as this work was initiated as part of translation of MAIAC's (Lyapustin et al., 2021) polarized radiative transfer solver IPOL (Korkin and Lyapustin, 2023) from FORTRAN into C."

In the very last paragraph of the Appendix, we made the following changes:

[original] "We are aware that the GRASP Team (https://www.grasp-open.com/) has also improved the DLS package. However, we have not evaluated their implementation yet."

[revised] "We are aware that the GRASP Team ( https://www.grasp-open.com/ ) has also improved the DLS package**: https://code.grasp-open.com/open/spheroid-package/ (simple registration is required)**. However, we have not evaluated their implementation yet **because results of their and our efforts were released simultaneously, about 3 years ago.**"

RESPONSE TO TECHICAL COMMENTS ABOUT SDLS:

"*It is not clear which translation is mentioned, was code manully re-written in C? FORTRAN and C share compliler and their translator makes same pseudocode for further compilation, this doesn't affect the speed of execution.*"

The Reviewer got it perfectly right (which also means, we put it in the right words): the code was manually re-written from FORTRAN into C. To emphasize that, we have reformulated the 1$^{st}$ bullet in the numbered list in the Appendix as explained above.

"*FORTRAN and C share compliler and their translator makes same pseudocode for further compilation, this doesn't affect the speed of execution.*"

AND

"*...FORTRAN and C binds naturally so the whole C translation for the performance looks a bit superficial.*"

We agree with the Reviewer and added the following sentence at the end of bullet 1: "**Note, however, that it is not the change of languages that contributes to numerical performance, but the code optimizations described below.**"

"*Majority of statements are either not directly related to the DLS package performance, but rather to the use case that was not optimal, ...*"

We do not fully agree with the reviewer here, assuming that by the "*statements*", the Reviewer means our bullets 1-4 in the SDLS Appendix. While bullet 1 (switching languages) is indeed not important for efficiency by itself, and mentioned only to explain why the translation was done (recall: for a different project, not relevant to the current manuscript), bullets 2 and 3 explicitly say what inconvenient (bullet 2: ASCII kernels vs. binary) or inefficient (bullet 3: 4 cubic spline interpolations, excessive for our simulations) are directly relevant to performance. The last bullet 4 (we separated the fixed kernels generating code from the one calculating the optical characteristics from the fixed kernels) is, like 1, does not change the performance. But it makes the code much shorter and transparent, and therefore easier to understand and support. For instance, the "*reading could be done only once per large retrieval sample*" option for the fixed kernels is much easier for implementing in the revised SDLS as compared to the original DLS because SDLS is much shorter, transparent and easier to understand - especially for non-developers.

"*It looks like the package wasn't used in the optimal way, and instead of changing several parameters in the it was re-written...*"

Strictly speaking, the legacy DLS code was organized in a way that does not match OUR needs for the speed of computations. SDLS fixes that. Other tasks may have different optimal ways.

*"it is not clear why loading kernels was such an issue, since they can be loaded once and then every-minute retrieval be performed with all the matrices already loaded."*

True, the code can be organized this way. However, a) the original DLS package does NOT do it; b) our retrieval algorithm calls the spheroidal package (previously DLS, now SDLS) and RT code as precompiled executables. Once a case run is over, the executable stops and RAM sets free. Therefore, the option "*all reading could be done only once per large retrieval sample*" is not available to us without major refactoring of the entire MFRSR retrieval algorithm. For multiple read-ins, we go with the binary format, which "*is more practical and faster*".

*"if compared to radiative transfer computational efforts, kernel reading and even interpolations shouldn't be such a performance issue…"*

That was our expectation too both in the current project as well as another Dr. Korkin's work (MAIAC by Lyapustin et al., 2021 – referenced in the manuscript) where he uses spheroids in combination with his own polarized RT code. But the execution time showed that the legacy DLS package was approximately as slow as multiple scattering simulations.

Note that we have not checked which component contributes most to the acceleration: the use of the binary files instead of ASCII, or dropping out 4 cubic spline interpolations, but the latter being a time-consuming routine.

" … *and due to this transformation* [of the fixed kernels from the original ASCII format to the binary one] *(which to my understanding is not completely justified, see above) these can't be automatically compared."*

We tested the new version of the package vs. the original legacy one by direct comparison of the packages outputs for a set of inputs (two reproducible examples are available from the /doc/ section which the Reviewer has noted) and by comparing the results of retrievals using the algorithm from the paper and running it first with the legacy then with the modified code.

*"I encourage authors to make the coding contributions more transparent and suitable for automatic affiliation research. Ideally, publish the code that converted kernels to binaries with proper link to the original kernel repository."*

We added source files into a new folder, /convert_kernels_src_linux/, and a step-by-step instruction (see readme.txt therein) on how to run the source and independently reproduce our numerical example. The original kernels in ASCII format are also provided in the same folder. In the second from the last paragraph, we added the following new sentence:

"The code for converting kernels to binaries is located in the mentioned GitHub's folder /convert_kernels_src_linux/ with step-by-step instructions and a reproducible example."

---

## Author Comment (AC2)

**Ground-based MFRSR UV-Vis spectral retrievals of Saharan dust absorption at Izaña Observatory**

Hiren Jethva, Nick Krotkov, Omar Torres, Jungbin Mok, Gordon Labow, Elena Lind, Tom Eck, Wei Gao, George Janson, Scott Simpson, Darrin Sharp, Kathy Lantz, Charles Wilson, Africa Barreto, Rosa García, Sergey Korkin, David Flittner

**Response to Referee # 2**

*\*Reviewer's comments: In blue-italic fonts*

**\*Authors' comments: In black-regular fonts**

*For the (non) sphericity part, the 1st reviewer has been asking a number of questions so I will not repeat some of them here.*

We encourage the referee to refer to our response to the comments related to the non-sphericity of dust aerosols offered by Referee # 1.

*Probably minor: Radiatively equivalent wavelength: Should this be different for Rayleigh, aerosol and gases since this is actually a multiplication of the filter function and the spectral function of each one of them, in the end integrated?*

Radiatively equivalent wavelengths are determined based on the spectral response function of each MFRSR filter, as shown in Figure 2 of the paper. All radiative transfer calculations for the Rayleigh, aerosols, and trace gases are carried out at the respective radiatively equivalent wavelengths for each filter.

*Methodology for UV retrievals. I think that the comparison of retrieved SSAs for 440nm from MFRSR and CIMEL is quite good and helps understanding that the method works as this "direct to global inversion"-based method agrees with a radiance based one. However, for the UV there are some additional aspects, that also in some extend are interconnected:*

*a. The CIMEL AOD@UV uncertainty is higher compared with the one at 440nm*

*b. The forward scattering effect on the field of view is also higher*

*c. There is an Angstrom based interpolation/extrapolation in order to retrieve the AOD at UV.*

The extrapolation of AERONET AOD to the shorter UV wavelengths of 325 nm and 332 nm is carried out following a quadratic relationship between AOD and wavelengths in log-log space using the 340-500 nm spectral AOD. Refer to section 3.1.

*d. There is a correction, based on a calibration constant derived on a different day*

The calibration constant ($lnV0$) of MFRSR is calculated on cleaner days with AOD440<0.1 and applied to the dusty days (AOD440>0.2) within the same month. In other words, the calibration constant correction is applied on monthly basis.

*Concerning a and c. It would be interesting to calculate the effect of CIMEL AOD UV uncertainty to the overall uncertainty of the final SSA calculations.*

The response to this comment is provided below.

*Combined a) with c): This uncertainty can impact a lot the calculated Angstrom exponents especially in dust related cases. For example: the paper example on figure 8 shows negative Angstrom (panel a) and almost zero Angstrom for b and c.  How uncertain this could be? and what would be the effect on SSA retrieval?*

The spectral AOD values reported in Figure 8 (a, b, c) are <daily averaged> values. The Ångström Exponent calculated from the 440-870 nm wavelength pair of AERONET for individual measurements might be different, i.e., positive or negative. While the Ångström Exponent is used to separate dust cases (AE<0.6), it is not used in the inversion procedure. The uncertainties in the retrieved SSA and imaginary part of the refractive index caused by error in the AOD measurements are added to the revision. Please refer to one of the following responses.

*Concerning the forward scattering effect: CIMEL is slightly affected but spectrally going down to 325nm (extrapolation) this could play a role.*

The 340 nm and 380 nm AOD uncertainty of ±0.02 in AERONET is dominated by calibration uncertainty and also interference filter degradation in time which impacts the calibration of field instruments. The effect of forward scattering in the CIMEL instrument FOV is not accounted for in this estimate. However, this effect is very small for low AOD days considered for deriving calibration constant for MFRSR. Uncertainties in retrieved $k$ and SSA for higher AOD days (AOD440>0.4) are quantified accounting for AERONET AOD uncertainty of ±0.02 at the UV wavelengths. Any further errors in the AERONET AOD arising from forward scattering would produce larger errors in the aerosol absorption retrieval than that estimated and described in Section 4: Uncertainty Characterization.

*In general looking for systematic effects e.g. an underestimation of the AOD (extrapolation, field of view forw. Scattering effect, etc) at equation 2 and through step2 (2.2.2) and 3 (2.2.3) will lead to a higher diffuse vs direct ratio (e.g. direct overestimation will lead also to diffuse underestimation and this will have an impact on the ratio). This can have a systematic effect for the Inversion part.  Maybe will be interesting to do a sensitivity analysis on such effect and the final outcome towards calculating SSA.*

Thank you for this important suggestion. We have reprocessed the entire Izaña MFRSR observational record added an estimate of the uncertainties in the retrieved imaginary part of the refractive index ($k$) and the single-scattering albedo (SSA) for each 1-minute measurement. The analysis explicitly accounts for the systematic uncertainties in the AERONET AOD (±0.01 at 440 nm and ±0.02 at UV wavelengths), as well as random uncertainties caused by 1% error in the MFRSR diffuse-to-direct irradiance measurements.

To address these points, we have added a new Section 4: Uncertainty Characterization, describing the methodology and resulting uncertainty estimates in detail. In addition, two new figures (Figures 7 and 8)

have been included, illustrating the dependence of the estimated uncertainties in $k$ and SSA on AOD, respectively.

*Minor comment: I think parts of lines 114-120 and 145-154 are similar so something can be erased or summarized.*

The content presented in section 2.1 is now transferred to the Introduction section, as it made more sense to discuss about the instrument and its deployment in the Introduction and prior to discussing the calibration procedure. Header structure and numbers for Section 2 are changed accordingly.